# MiniX: Mitigating Low-Rank Collapse and Attention Bottlenecks in Tabular Foundation Models

## Abstract

Recent tabular foundation models routinely match or surpass strong tree ensembles and specialized deep architectures, yet their numeric embeddings remain a bottleneck. We diagnose a low-rank collapse induced by the prevalent linear+ID scheme and introduce RaBEL, a compact Radial Basis Embedding Layer that front-loads nonlinearity via localized RBF features. RaBEL increases shallow-layer effective rank and improves conditioning without deeper stacks; it is complementary to periodic mappings. We further identify a permutation-order pathology in bidirectional attention (feature→sample) and propose a reordered stack: sample-attention → FFN → feature-attention, ensuring column-level context precedes feature mixing and that all attention computations influence the readout. Combining both ideas yields MiniX, a 2M-parameter model that surpasses 7M-parameter TabPFN-v2 and 27M-parameter TabICL baselines on popular benchmarks while reducing training and inference cost. Our results highlight principled nonlinear embeddings and attention-order redesign as key enablers of accuracy and efficiency gains in tabular foundation models.

## 1 Introduction

Recent advances in tabular foundation models—most notably TabPFN/TabPFN-v2 (Hollmann et al., 2025b; 2022), TabICL (Qu et al., 2025), and LimiX (Zhang et al., 2025), have narrowed, and in many regimes surpassed, long-standing baselines such as gradient-boosted decision trees (Chen & Guestrin, 2016; Ke et al., 2017; Prokhorenkova et al., 2018) and specialized deep tabular architectures (Somepalli et al., 2021; Arik & Pfister, 2021). These results are striking given the historical dominance of tree ensembles on medium-scale tabular tasks and the difficulty deep models have shown on irregular functions, heavy tails, and mixed data types (Grinsztajn et al., 2022).

Despite this progress, the input embedding layer remains a central limitation. The prevailing recipe maps each scalar cell through a single linear layer and augments it with a column identifier (e.g., positional or feature-ID embeddings), as in TabTransformer and FT-Transformer (Huang et al., 2020; Gorishniy et al., 2021). Systematic analysis indicates that such numeric embeddings are often overly restrictive and leave substantial performance on the table relative to more expressive encodings (Gorishniy et al., 2022). In our profiling, this design induces highly correlated activations early in the network: feature matrices in shallow layers can exhibit extremely low effective rank, sometimes collapsing to single-digit ranks on common benchmarks. This phenomenon implies significant parameter redundancy, suggesting that comparable performance could be achieved with a much smaller parameter budget. Moreover, it highlights untapped representational capacity: by rectifying this rank collapse to fully utilize the latent space, there is potential to unlock substantially richer feature representations.

We argue that the embedding layer for tabular FMs should play a greater role in introducing nonlinearity, enabling early representations to separate common tabular phenomena such as piecewise trends, local periodicity, quantization, heavy-tailed marginals, and heteroskedasticity. To this end, we propose RaBEL—a Radial Basis Embedding Layer—that replaces the one-shot linear projection with a bank of localized nonlinear features. Classical theory and practice support RBF features as universal, localized approximators closely connected to kernel methods (Broomhead & Lowe, 1988;

Park & Sandberg, 1991; Schölkopf & Smola, 2002; Rasmussen & Williams, 2006). Compared to direct linear embeddings, the localized nature of RBFs (i) yields diverse activation patterns across value regimes, (ii) improves conditioning of the first learned layer, and (iii) raises the effective rank of shallow representations without requiring many stacked layers to "discover" curvature ex post. The approach is complementary to periodic/Bochner-style mappings (e.g., random Fourier features) and can be extended or hybridized when periodicity is expected (Rahimi & Recht, 2007).

Beyond embeddings, we identify a second limitation in the permutation order of bidirectional attention used by existing tabular foundation models. In widely used designs (e.g., TabPFN-style or LimiX-style stacks), attention is typically arranged as feature-attention → sample-attention. This ordering introduces two problems. **(1)** In the very first layer, feature-level attention must integrate across columns based solely on raw values, before any column-level statistics or correlations have been established; this deprives attention of informative context and exacerbates low-rank collapse. **(2)** During prediction, many architectures consume only the target token from the final layer; intermediate feature embeddings—and thus the sample-level attention computed over features—are effectively ignored, leading to weak training signals for parts of the network that do not directly influence the readout.

We address these issues by reordering the attention stack to sample-attention → (FFN) → feature-attention. The sample-attention phase at the input stage allows the model to aggregate column-level correlations and distributional statistics (e.g., moments, prevalence, missingness patterns) before engaging feature-level attention. An intermediate feed-forward network (FFN) then compresses and conditions these signals, after which the feature-attention phase learns inter-feature relations using richer, better-conditioned inputs. This permutation ensures that all attention computations contribute to the final prediction: information assembled at the sample-level directly shapes the feature-level representations that flow to the readout. This refined mechanism improves the capture of feature relationships and encourages the model to discover critical features, a capability we further analyze in the Section A.3.

With the combination of RaBEL and the reordered sample-attention → FFN → feature-attention architecture, we introduce a 2 M-parameter model, named **MiniX**, that surpasses the 7 M-parameter TabPFN-v2 baseline on mainstream benchmarks while cutting computational costs for both training and inference. These results indicate that principled nonlinear embeddings coupled with attention-order redesign can unlock better accuracy–efficiency trade-offs and more reliable scaling for tabular foundation models.

We summarize our contributions as follows.

1. We diagnose and quantify the low-rank collapse induced by linear+ID embeddings, and propose RaBEL, a compact RBF-based cell encoder that raises shallow-layer rank and improves conditioning.

2. We reveal a permutation-order pathology in standard bidirectional attention (feature-attention → sample-attention), and introduce a sample-attention → FFN → feature-attention stack that (i) establishes column-level statistics before feature aggregation and (ii) routes all attention signals to the readout.

3. We introduce MiniX, a 2M-parameter model building on these two components that outperforms TabPFN-v2 with 7M-parameter and TabICL with 27M-parameter on most benchmarks while reducing both training and inference cost.

## 2 RELATED WORK

**Deep models for tabular data.** Gradient-boosted decision trees (GBDTs) such as XGBoost, LightGBM, and CatBoost have long dominated tabular prediction (Chen & Guestrin, 2016; Ke et al., 2017; Prokhorenkova et al., 2018). In response, specialized neural architectures have been proposed. TabNet performs attentive feature selection with interpretability (Arik & Pfister, 2021), TabTransformer applies self-attention over features—especially effective for categorical inputs (Huang et al., 2020), and SAINT augments feature-wise attention with intersample (row-wise) attention and contrastive pre-training (Somepalli et al., 2021). Despite these advances, broad evaluations still often find trees competitive on medium-scale benchmarks (Grinsztajn et al., 2022), underscoring the

challenges of mixed data types and irregular target functions and motivating foundation-style approaches.

**Tabular foundation models.**   TabPFN reframes tabular learning as in-context inference: a Transformer pre-trained on synthetic tasks consumes a small training set at inference and predicts without gradient updates (Hollmann et al., 2022; 2025b). TabICL adopts a two-stage design that first builds per-sample representations with feature-then-row attention, followed by efficient in-context reasoning (Qu et al., 2025). LimiX treats a table as a joint distribution over features and missingness and uses masked modeling to support many tasks in one model (Zhang et al., 2025). Contemporary systems such as Mitra (Zhang & Danielle, 2025) explore hybrid row–column attention with synthetic priors to improve cross-dataset generalization.

**Embedding strategies for numerical features.**   A key design choice is how to encode continuous-valued cells. A prevalent recipe applies a single linear projection per numeric column, sometimes with column-identity embeddings, as in FT-Transformer and TabTransformer (Gorishniy et al., 2021; Huang et al., 2020). Systematic analyses show that such encodings can be restrictive relative to more expressive schemes (Gorishniy et al., 2022). Effective remedies include (i) piecewise-linear encodings that partition value ranges into learnable segments and (ii) periodic encodings via sinusoidal features, conceptually related to random Fourier features (Gorishniy et al., 2022; Rahimi & Recht, 2007). Complementary self-supervised pre-training (e.g., VIME, MET) uses masked reconstruction to capture inter-feature dependencies prior to supervised fine-tuning (Yoon et al., 2020; Majmundar et al., 2022). Kernel-inspired alternatives based on radial basis functions (RBFs) offer localized receptive fields and universal approximation (Broomhead & Lowe, 1988; Park & Sandberg, 1991), closely connected to Gaussian RBF kernels and Gaussian processes (Schölkopf & Smola, 2002; Rasmussen & Williams, 2006). Learned RBF featurization thus provides localized nonlinear transformations that complement global periodic mappings like random Fourier features and can be naturally integrated into transformer-based tabular backbones.

## 3   THE LOW RANK PROBLEM IN CURRENT EMBEDDINGS

The current embedding strategy adopted by TabPFN-v2 is simply mapping each cell, i.e. each scalar, to the high-dimensional hidden space via a $1 \times p$ linear projection. Such a straightforward strategy implicitly leads to the low-rank problem: The hidden states output by transformer layers tend to be low-rank, especially early in the network. This could severely decrease the expressivity of the network, leading to potential performance degradation. In this section, we conduct both theoretical and experimental analyses to reveal the low-rank problem in TabPFN-v2.

First we theoretically prove the existence of the low-rank problem.

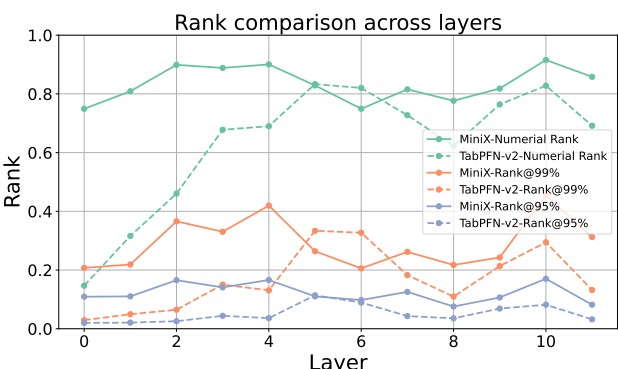

Figure 1: Rank comparison across layers of MiniX and TabPFN-v2. The metric Rank@99% and Rank@95% represents the minimum number of SVD components required to take up 99% or 95% energy measured by singular values.

**Proposition 3.1** *Assume the linear embedding strategy without positional embeddings, given a scalar sequence $x_1, x_2, ..., x_n \in \mathbb{R}$, the rank of the embedded input matrix is at most 2. For standard self-attention, the rank of the output matrix is at most 2. For multi-head attention with the number of heads $H$, the rank of the output is at most $H + 1$.*

The detailed proof can be referred to in Appendix A.1. Proposition 3.1 reveals that under the linear embedding strategy, the embedded input and the hidden states of shallower transformer layers could be extremely low-rank if no other non-linearity is induced. This is highly inefficient for such a high-dimensional hidden space.

For empirical analyses, we calculate the average rank of hidden states output by each layer across datasets in OpenML-CC18. For the hidden states of $X$, we flatten the two dimensions of samples and features into one dimension so that it turns into a 2D matrix. In Figure 1, the numerical rank represents the matrix rank calculated via the off-the-shelf numpy function where a threshold is set to filter the singular values to determine the rank after singular value decomposition (SVD). The metric Rank@99% and Rank@95% represent the minimum number of components required to take up 99% or 95% energy measured by singular values. All the metrics are normalized by the hidden dimension size. We can see that hidden states of TabPFN-v2 are generally low-rank, while our model MiniX increases the rank by a large margin in the first five layers.

To verify the low-rank problem in the TabPFN-v2 model, we applied low-rank approximation to the feature-sample embedding matrix during each forward pass. The experiments show that even with the rank reduced significantly (e.g., to $\approx 1/10$ of the hidden dimension), the performance (AUC) remains competitive, indicating highly inefficient utilization of the model's hidden space and the need for a better embedding strategy. See details in Section A.2.

## 4 METHOD

### 4.1 PROBLEM SETUP AND NOTATION

Let $X \in \mathbb{R}^{N \times D}$ denote a tabular dataset with $N$ samples (rows) and $D$ features (columns). We write $x_{i,j}$ for the scalar value at sample $i$ and feature $j$. Our goal is to map each cell $x_{i,j}$ to an embedding $e_{i,j} \in \mathbb{R}^d$ that is fed into a transformer with bidirectional attention operating along the sample and feature axes. We denote by $E \in \mathbb{R}^{N \times D \times d}$ the full tensor of cell embeddings, and by $L$ the number of attention blocks in the backbone.

This section introduces (i) a compact radial-basis embedding layer, **RaBEL**, that replaces the standard linear projection for scalar cells, and (ii) a reordered bidirectional attention block, from **Feature-Attention → Sample-Attention → FFN** (abbrev. F→S→N) to **Sample-Attention → FFN → Feature-Attention** (abbrev. S→N→F), that improves conditioning and ensures all attention computations contribute to the final prediction.

### 4.2 RABEL: RADIAL-BASIS EMBEDDING LAYER

Direct linear projection of numeric cells produces highly correlated early activations and very low effective rank in the first layers, which inhibits downstream capacity. RaBEL front-loads nonlinearity by expanding each scalar into a small bank of localized responses, followed by a light projection to the model dimension.

**Per-column normalization.** For numerical stability we standardize each column using the z-score:

$$\tilde{x}_{i,j} \;=\; \frac{x_{i,j} - \bar{x}_j}{s_j + \epsilon}, \qquad \bar{x}_j \;=\; \frac{1}{N}\sum_{i=1}^{N} x_{i,j}, \qquad s_j \;=\; \sqrt{\frac{1}{N-1}\sum_{i=1}^{N}\left(x_{i,j} - \bar{x}_j\right)^2}.$$

Here, $\bar{x}_j$ and $s_j$ are the per-column sample mean and standard deviation, respectively, and $\epsilon > 0$ is a small constant for numerical stability.

All formulas below apply to either $x_{i,j}$ or $\tilde{x}_{i,j}$; we drop the tilde for brevity.

**RBF expansion.** For each feature $j$ we choose $M$ centers $\{c_{j,m}\}_{m=1}^{M}$ and bandwidths $\{\sigma_{j,m}\}_{m=1}^{M}$. Centers are initialized at empirical quantiles of $X_{\cdot,j}$; bandwidths are initialized proportional to local variability (e.g., interquartile range), and all parameters are subsequently learned end-to-end. The radial-basis feature map is

$$\phi_j(x_{i,j}) \;=\; \left[\, \exp\!\left(-\tfrac{(x_{i,j}-c_{j,1})^2}{2\sigma_{j,1}^2}\right),\, \ldots,\, \exp\!\left(-\tfrac{(x_{i,j}-c_{j,M})^2}{2\sigma_{j,M}^2}\right)\right] \in \mathbb{R}^M.$$

**Projection to model dimension.** A shared linear projection with LayerNorm maps the expanded features to the model width:

$$e_{i,j} \;=\; \mathrm{LN}\big(W_{\mathrm{rbf}}\,\phi_j(x_{i,j}) + b_{\mathrm{rbf}}\big) \in \mathbb{R}^d,$$

where $W_{\mathrm{rbf}} \in \mathbb{R}^{d\times M}$ and $b_{\mathrm{rbf}} \in \mathbb{R}^d$ are learned and shared across columns (unless otherwise stated). For categorical columns we use a standard entity embedding lookup into $\mathbb{R}^d$, followed by the same LayerNorm.

**Exponent-Gated RaBEL with *shared gates* (scale-conditioned modulation).** Real-world tabular values span orders of magnitude, may be reported in different units, and often exhibit heteroskedasticity. To make RaBEL *scale-aware* without losing locality, we introduce an **exponent gate** that conditions the *parameters* of the RBF bank via *shared* scalar gates applied uniformly across all $M$ basis functions for the current cell.

*Exponent extraction (soft, differentiable).* Fix a base $\beta > 1$ (we use $\beta = 2$) and a small offset $\tau > 0$. Define the log-magnitude

$$\ell_{i,j} \;=\; \log_\beta\big(|x_{i,j}| + \tau\big).$$

Let $\mathcal{B} = \{b_{\min}, \ldots, b_{\max}\} \subset \mathbb{Z}$ be a bounded set of exponent bins. We form a soft assignment to exponent bins via a temperature-controlled kernel:

$$\pi_{i,j}(b) \;=\; \frac{\exp\big(-(\ell_{i,j}-b)^2/T\big)}{\sum_{b'\in\mathcal{B}}\exp\big(-(\ell_{i,j}-b')^2/T\big)}, \qquad b \in \mathcal{B},\ T > 0.$$

Each bin $b$ has a learnable embedding $u_b \in \mathbb{R}^h$, and we also include a sign embedding $u_{\mathrm{sgn}(x_{i,j})} \in \mathbb{R}^{h_s}$ for $\mathrm{sgn}(x) \in \{-1, 0, +1\}$. We obtain the *scale context* by concatenating ( $\|$ ) the weighted bin embedding and the sign embedding:

$$z_{i,j}^{\mathrm{exp}} \;=\; \Big(\sum_{b\in\mathcal{B}} \pi_{i,j}(b)\, u_b\Big) \;\|\; u_{\mathrm{sgn}(x_{i,j})} \;\in\; \mathbb{R}^{h+h_s}.$$

*Shared gates on* $(c, \sigma)$. A tiny MLP produces two positive scalars per cell that are *shared* across the entire RBF bank:

$$[\gamma_{i,j}^c,\, \gamma_{i,j}^\sigma] \;=\; \mathrm{MLP}_{\mathrm{shared}}(z_{i,j}^{\mathrm{exp}}) \in \mathbb{R}^2, \qquad \gamma_{i,j}^c, \gamma_{i,j}^\sigma \;>\; 0,$$

where positivity is enforced via an exponential or softplus transform. We compute exponent-conditioned centers and widths

$$c_{j,m}^{(\mathrm{exp})}(i) \;=\; \gamma_{i,j}^c\, c_{j,m}, \qquad \sigma_{j,m}^{(\mathrm{exp})}(i) \;=\; \gamma_{i,j}^\sigma\, \sigma_{j,m},$$

and evaluate the gated RBF features

$$\phi_j^{(\mathrm{exp})}(x_{i,j}) \;=\; \Big[\, \exp\big(-\tfrac{(x_{i,j}-c_{j,1}^{(\mathrm{exp})}(i))^2}{2(\sigma_{j,1}^{(\mathrm{exp})}(i))^2}\big),\, \ldots,\, \exp\big(-\tfrac{(x_{i,j}-c_{j,M}^{(\mathrm{exp})}(i))^2}{2(\sigma_{j,M}^{(\mathrm{exp})}(i))^2}\big) \Big],$$

followed by the same projection

$$e_{i,j} \;=\; \mathrm{LN}\big(W_{\mathrm{rbf}}\phi_j^{(\mathrm{exp})}(x_{i,j}) + b_{\mathrm{rbf}}\big).$$

Exponent gating brings three benefits: (i) **Scale equivariance**: multiplying inputs by $\beta$ shifts $\ell$ by 1, thus the gate adapts the RBF bank across orders of magnitude, yielding unit- and scale-robust embeddings; (ii) **Heteroskedasticity robustness**: widths and amplitudes expand/contract in high/low variance regimes, maintaining useful locality of the bumps; (iii) **Better conditioning**: separating magnitude (via the exponent pathway) from pattern within a decade (via RBF responses) produces higher effective rank and smoother gradients in early layers. The soft assignment $\pi_{i,j}$ makes the module fully differentiable and avoids brittle hard binning. The computational overhead is small: an extra $O(|\mathcal{B}|)$ kernel evaluation and a tiny two-layer MLP.

### 4.3 Reordered Bidirectional Attention

Tabular transformers bidirectional-attention typically alternate attention across features (per sample) and across samples (per feature). Common stacks has the order of **FSN**(feature-attention → sample-attention → feed-forward). This forces the initial feature-level attention to integrate across columns using raw, weakly conditioned values. Moreover, the final prediction consumes only the target token, leaving other feature embeddings and thus the last sample-level attention computations underutilized.

We modify the bidirectional block by reordering its modules—without introducing any additional components—to **SNF**(sample-attention → feed-forward → feature-attention). In this design, the sample-attention layer first aggregates column-level statistics and cross-sample regularities, a lightweight feed-forward network conditions these signals, and the feature-attention layer then models inter-feature relations on better-conditioned inputs. The final embedding for prediction is obtained via attention pooling over all feature tokens, so every attention computation contributes directly to the output.

## 5 Experiments

### 5.1 Embedding Comparison

#### 5.1.1 MLP-based Methods

We benchmark RaBEL against four baselines: No-embedding, MLP, PLE, and Periodic across diverse datasets (Table 1). For the embedding methods, we adopt a unified formulation where each scalar input $x_{j,i}$ is first transformed by a function $\phi_\theta : \mathbb{R} \to \mathbb{R}^d$, then flattened and mapped to the final dimension $e$ via a linear layer. The methods differ solely in $\phi_\theta$: **MLP** employs a pointwise FFN; **PLE** and **Periodic** utilize piecewise-linear and sinusoidal encodings (Gorishniy et al., 2022), respectively. Note that we modify PLE and Periodic to use *shared* linear projections to map encoded values to $\mathbb{R}^d$, rather than feature-specific ones. This adaptation accommodates varying feature counts, facilitating their subsequent application in foundation models. The **No-embedding** baseline directly projects the raw input $\mathbf{x}$ to $\mathbb{R}^{s \times e}$. Results are detailed in Table 2.

Table 1: Statistics of datasets. These datasets cover varying sample sizes, tasks, data distributions, and levels of missingness. AUC: area under the ROC curve; Acc.: classification accuracy; $R^2$: coefficient of determination; RMSE: root mean squared error.

| Dataset | GC | CP | AC | CB | UK | BN | MA | CD | MH | CC | HS | SC | MV |
|---|---|---|---|---|---|---|---|---|---|---|---|---|---|
| #train samples | 205 | 696 | 7000 | 5455 | 282 | 44236 | 30305 | 726 | 9643 | 671 | 15129 | 36421 | 28537 |
| #features | 9 | 3 | 7 | 95 | 5 | 9 | 4 | 12 | 15 | 8 | 20 | 4 | 10 |
| #cate features | 7 | 1 | 5 | 2 | 0 | 7 | 0 | 2 | 2 | 0 | 4 | 1 | 3 |
| #Metric1 | AUC | AUC | AUC | AUC | AUC | AUC | AUC | $R^2$ | $R^2$ | $R^2$ | $R^2$ | $R^2$ | $R^2$ |
| #Metric2 | Acc. | Acc. | Acc. | Acc. | Acc. | Acc. | Acc. | RMSE | RMSE | RMSE | RMSE | RMSE | RMSE |
| #classes | 7 | 4 | 2 | 2 | 5 | 3 | 2 | – | – | – | – | – | – |
| #missing ratio | 0.2366 | 0 | 0.2892 | 0 | 0 | 0 | 0 | 0.00056 | 0 | 0 | 0 | 0 | 0 |

Table 2: Results for MLP with different embedding modules. The upper panel reports results for Metric1, and the lower panel reports results for Metric2. For each dataset, the arrow indicates whether higher or lower values are better for the corresponding metric.

| Metric1 | GC (↑) | CP (↑) | AC (↑) | CB (↑) | UK (↑) | BN (↑) | MA (↑) | CD (↑) | MH (↑) | CC (↑) | HS (↑) | SC (↑) | MV (↑) |
|---|---|---|---|---|---|---|---|---|---|---|---|---|---|
| MLP | 0.7537 | 0.8092 | 0.6560 | 0.8319 | 0.9850 | 0.7520 | **0.8896** | 0.4476 | 0.7811 | 0.6158 | 0.7497 | 0.1799 | **0.9999** |
| MLP-MLP | 0.8984 | 0.8496 | 0.8398 | 0.8924 | **0.9863** | 0.7518 | 0.8577 | 0.4768 | 0.8618 | 0.8947 | 0.8375 | 0.1749 | 0.9990 |
| MLP-PLE | 0.7393 | 0.8031 | 0.8262 | 0.8817 | 0.8416 | 0.7389 | 0.8609 | 0.2969 | 0.8281 | 0.8576 | 0.8279 | **0.1815** | 0.9730 |
| MLP-Periodic | 0.7817 | 0.8895 | 0.8054 | 0.8926 | 0.9821 | **0.7586** | 0.8625 | 0.4095 | 0.8306 | **0.9068** | 0.8286 | 0.1786 | 0.9994 |
| MLP-RaBEL | **0.9831** | **0.9582** | **0.9061** | **0.8979** | 0.9677 | 0.7580 | 0.8789 | **0.5124** | **0.8818** | 0.8998 | **0.8401** | 0.1813 | 0.9995 |

| Metric2 | GC (↑) | CP (↑) | AC (↑) | CB (↑) | UK (↑) | BN (↑) | MA (↑) | CD (↓) | MH (↓) | CC (↓) | HS (↓) | SC (↓) | MV (↓) |
|---|---|---|---|---|---|---|---|---|---|---|---|---|---|
| MLP | 0.3483 | 0.6756 | 0.6750 | 0.9677 | **0.9008** | 0.5757 | 0.9611 | 0.7218 | 0.4862 | 0.6072 | 0.5259 | 0.9051 | **0.0119** |
| MLP-MLP | 0.5393 | 0.7224 | 0.7870 | **0.9692** | **0.9008** | 0.5766 | 0.9657 | 0.7025 | 0.3863 | 0.3178 | 0.4238 | 0.9078 | 0.0321 |
| MLP-PLE | 0.2584 | 0.6890 | 0.7750 | 0.9677 | 0.5868 | 0.5596 | 0.9654 | 0.8143 | 0.4308 | 0.3697 | 0.4361 | **0.9042** | 0.1642 |
| MLP-Periodic | 0.3371 | 0.7759 | 0.7637 | 0.9677 | 0.8595 | **0.5788** | 0.9666 | 0.7463 | 0.4277 | **0.2991** | 0.4353 | 0.9058 | 0.0246 |
| MLP-RaBEL | **0.8090** | **0.8495** | **0.8377** | **0.9692** | 0.8678 | 0.5758 | **0.9668** | **0.6781** | **0.3573** | 0.3101 | **0.4203** | 0.9043 | 0.0221 |

### 5.1.2 TRANSFORMER-BASED METHODS

We integrate different embedding methods into a 2M-parameter transformer backbone with the same settings as in the previous section. Following the foundation-model training paradigm (training on generated data), we evaluate them on BCCO-CLS and BCCO-REG (Zhang et al., 2025). Results can be found in Table 3 and Table 4.

Table 3: Results on BCCO-CLS with different embedding modules.

| Embedding | AUC (↑) | Acc. (↑) | F1 (↑) |
|---|---|---|---|
| Transformer+MLP | 83.52 | 76.82 | 66.57 |
| Transformer+Periodic | 83.88 | 77.80 | 68.65 |
| Transformer+PLE | 84.66 | 77.68 | 67.74 |
| Transformer+RaBEL | **85.04** | **77.99** | **69.01** |

Table 4: Results on BCCO-REG with different embedding modules.

| Embedding | $R^2$ (↑) | RMSE (↓) |
|---|---|---|
| Transformer+MLP | 0.7731 | 0.4043 |
| Transformer+Periodic | 0.6859 | 0.4321 |
| Transformer+PLE | 0.7410 | 0.4216 |
| Transformer+RaBEL | **0.7792** | **0.3964** |

Table 5: Classification results on **TabArena-CLS**, sorted by mean AUC in descending order, with the parameter counts of all foundation models highlighted in blue.

| Model | TabArena | | | | | |
|---|---|---|---|---|---|---|
| | Mean | | | Rank | | |
| | AUC (↑) | Acc. (↑) | F1 (↑) | AUC (↓) | Acc. (↓) | F1 (↓) |
| LimiX (16.52M) | 0.849 | 0.877 | 0.597 | 3.636 | 3.424 | 7.273 |
| MiniX (1.92M) | 0.846 | 0.876 | 0.594 | 5.000 | 3.394 | 6.909 |
| AutoGluon | 0.844 | 0.870 | 0.574 | 5.909 | 5.606 | 9.545 |
| TabICL (27.10M) | 0.840 | 0.870 | 0.553 | 7.182 | 6.636 | 11.000 |
| TabPFN-v2 (7.24M) | 0.838 | 0.872 | 0.589 | 7.485 | 4.697 | 9.152 |
| LightGBM | 0.841 | 0.868 | 0.574 | 7.606 | 8.606 | 10.970 |
| XGBoost | 0.838 | 0.867 | 0.567 | 8.545 | 7.970 | 11.273 |
| RF | 0.837 | 0.864 | 0.558 | 10.061 | 8.697 | 12.545 |
| CatBoost | 0.835 | 0.867 | 0.574 | 10.273 | 7.818 | 10.242 |
| ET | 0.833 | 0.857 | 0.505 | 11.212 | 11.515 | 16.879 |
| Mitra (75.67M) | 0.815 | 0.862 | 0.533 | 12.667 | 10.636 | 15.545 |
| TabM | 0.807 | 0.855 | 0.516 | 15.212 | 15.212 | 13.970 |
| ExcelFormer | 0.810 | 0.849 | 0.555 | 15.455 | 17.485 | 15.515 |
| RealMLP | 0.809 | 0.751 | 0.477 | 17.303 | 25.121 | 18.061 |
| MLP | 0.772 | 0.822 | 0.459 | 19.212 | 18.212 | 20.212 |
| TANGOS | 0.791 | 0.844 | 0.522 | 19.455 | 18.364 | 16.576 |
| T2G-Former | 0.779 | 0.822 | 0.482 | 19.697 | 20.576 | 16.273 |
| MLP-PLR | 0.781 | 0.836 | 0.460 | 19.818 | 19.485 | 20.152 |
| ModernNCA | 0.783 | 0.846 | 0.511 | 20.576 | 21.061 | 17.667 |
| AutoInt | 0.769 | 0.826 | 0.474 | 20.636 | 21.121 | 19.182 |
| TabR | 0.785 | 0.842 | 0.510 | 21.061 | 20.182 | 16.545 |
| NODE | 0.769 | 0.792 | 0.352 | 21.182 | 21.939 | 24.970 |
| ResNet | 0.781 | 0.824 | 0.532 | 21.364 | 22.182 | 16.818 |
| FT-Transformer | 0.770 | 0.803 | 0.468 | 21.485 | 22.061 | 19.030 |
| TabTransformer | 0.739 | 0.781 | 0.438 | 21.667 | 21.970 | 19.606 |
| DCN-v2 | 0.769 | 0.833 | 0.482 | 22.333 | 19.909 | 18.606 |
| SNN | 0.755 | 0.818 | 0.442 | 23.242 | 22.545 | 22.545 |
| TabCaps | 0.742 | 0.837 | 0.471 | 23.273 | 18.515 | 20.212 |
| SwitchTab | 0.754 | 0.799 | 0.409 | 24.091 | 25.545 | 22.758 |
| DANets | 0.749 | 0.776 | 0.453 | 24.273 | 24.303 | 18.727 |
| SAINT | 0.694 | 0.739 | 0.437 | 25.485 | 25.758 | 22.121 |
| TabNet | 0.709 | 0.789 | 0.438 | 26.818 | 22.879 | 23.727 |
| GrowNet | 0.646 | 0.674 | 0.361 | 27.697 | 28.939 | 23.909 |

## 5.2 TOY EXPERIMENTS OF RBA

To empirically validate the capability of SNF in capturing latent feature dependencies, we conducted a toy experiment using a synthetic dataset generated from a Directed Acyclic Graph (DAG). Our analysis reveals that while the FSN baseline is dominated by self-attention, SNF accurately assigns

high attention scores to the direct causal features of the target, confirming its superior ability to model complex feature interactions. We refer readers to Section A.3 for the detailed experimental setup and visualization of attention maps.

Table 6: Regression results on **CTR23**, sorted by mean $R^2$ in descending order, with the parameter counts of all foundation models highlighted in blue.

| | CTR23 | | | |
| | Mean | | Rank | |
| Model | $R^2$ ($\uparrow$) | RMSE ($\downarrow$) | $R^2$ ($\downarrow$) | RMSE ($\downarrow$) |
|---|---|---|---|---|
| LimiX (16.52M) | 0.745 | 0.477 | 4.545 | 4.667 |
| AutoGluon | 0.725 | 0.497 | 5.939 | 5.848 |
| RealMLP | 0.721 | 0.494 | 6.333 | 6.303 |
| TabM | 0.719 | 0.494 | 6.515 | 6.545 |
| MiniX (1.92M) | 0.730 | 0.495 | 7.636 | 7.667 |
| TabPFN-v2 (7.24M) | 0.716 | 0.503 | 8.485 | 8.394 |
| XGBoost | 0.712 | 0.511 | 9.818 | 9.879 |
| LightGBM | 0.706 | 0.516 | 10.848 | 10.939 |
| ET | 0.697 | 0.535 | 11.939 | 12.000 |
| CatBoost | 0.700 | 0.528 | 12.394 | 12.394 |
| RF | 0.694 | 0.539 | 12.818 | 12.788 |
| ExcelFormer | 0.665 | 0.556 | 13.848 | 13.939 |
| T2G-Former | 0.674 | 0.544 | 15.061 | 15.030 |
| DCN-v2 | 0.670 | 0.545 | 15.788 | 15.848 |
| ResNet | 0.645 | 0.587 | 15.879 | 15.939 |
| FT-Transformer | 0.667 | 0.549 | 15.909 | 16.000 |
| TabR | 0.671 | 0.543 | 16.182 | 16.091 |
| MLP-PLR | 0.672 | 0.553 | 16.909 | 16.939 |
| ModernNCA | 0.667 | 0.550 | 17.091 | 16.788 |
| MLP | 0.608 | 0.623 | 17.121 | 17.121 |
| SAINT | 0.654 | 0.561 | 17.939 | 17.848 |
| Mitra (75.67M) | 0.624 | 0.583 | 17.939 | 18.061 |
| TANGOS | 0.642 | 0.586 | 18.121 | 18.152 |
| AutoInt | 0.655 | 0.568 | 18.970 | 18.939 |
| NODE | 0.568 | 0.666 | 21.697 | 21.545 |
| TabNet | 0.605 | 0.623 | 21.818 | 21.879 |
| DNNR | -2.969 | 1.651 | 23.909 | 23.909 |
| SNN | 0.369 | 0.834 | 26.758 | 26.788 |
| GrowNet | 0.185 | 0.944 | 28.667 | 28.636 |
| DANets | 0.001 | 1.052 | 30.000 | 30.000 |
| TabTransformer | 0.000 | 1.053 | 30.030 | 30.030 |
| SwitchTab | -0.006 | 1.057 | 31.091 | 31.091 |

## 5.3 Comparison With SOTA Models

**Training Setting.** We construct our pre-training corpus using hierarchical Structural Causal Models (SCMs), following the data generation protocols established in the PFN series (Hollmann et al., 2022; 2025a) and LimiX (Zhang et al., 2025). In each training episode, we first sample a random Directed Acyclic Graph (DAG) and functional mechanisms to define a specific joint distribution, from which synthetic data samples are subsequently drawn. The backbone of MiniX is a 12-block Transformer architecture designed to capture both inter-sample and intra-feature dependencies. Each block is distinctively composed of three components in sequence: a **Sample-Attention** module, a **Feed-Forward Network (FFN)**, and a **Feature-Attention** module. The Sample-Attention mechanism facilitates interaction across different data samples (rows), while the Feature-Attention mechanism models the relationships between variables (columns) within a sample. The model is configured with a hidden embedding dimension of $d_{\text{model}} = 96$ and employs $H = 6$ attention heads in each attention module, balancing computational efficiency with expressive power.

### 5.3.1 Classification

**Benchmarks.** We draw from six benchmark suites: TALENT-CLS (Liu et al., 2024), OpenML-CC18 (Bischl et al., 2017), PFN-CLS (Hollmann et al., 2025a), TabZilla (McElfresh et al., 2023),

Table 7: Ablation on RBA and RaBEL. We report AUC on **TabArena** and **TabZilla**.

| Model | RBA | RaBEL | TabArena-AUC ($\uparrow$) | TabZilla-AUC ($\uparrow$) |
|---|---|---|---|---|
| baseline | – | – | 0.8215 | 0.9180 |
| +RaBEL | – | ✓ | 0.8399 | 0.9285 |
| +RBA | ✓ | – | 0.8301 | 0.9293 |
| MiniX(RBA+RaBEL) | ✓ | ✓ | **0.8431** | **0.9313** |

TabArena-CLS (Erickson et al., 2025), and BCCO-CLS (Zhang et al., 2025). Following a common protocol, datasets with more than 50 000 training samples, over 10 000 features, or more than 10 target classes were excluded. After filtering, the final collection comprised 179 datasets from TALENT-CLS, 62 from OpenML-CC18, 29 from PFN-CLS, 27 from TabZilla, 33 from TabArena-CLS, and 106 from BCCO-CLS.

**Metrics.** We evaluated performance using three metrics—AUC (Area Under the ROC Curve), Acc. (Accuracy), and F1 (F1-score). For multi-class AUC and F1 calculations, we adopted a one-vs-one strategy.

### 5.3.2 REGRESSION

**Benchmarks.** We use five open-source benchmarks, TALENT-REG (Liu et al., 2024), PFN-REG (Hollmann et al., 2025a), TabArena-REG (Erickson et al., 2025), CTR23 (Fischer et al., 2023), and BCCO-REG (Zhang et al., 2025). Applying the same filtering criteria as for the classification benchmarks yields 33 datasets from CTR23, 28 from PFN-REG, 13 from TabArena-REG, 99 from TALENT-REG, and 50 from BCCO-REG.

**Metrics.** We assessed regression performance using two metrics: $R^2$ (coefficient of determination) and RMSE (root mean squared error).

### 5.3.3 RESULTS

Due to space constraints, we report additional results in the Section A.5. As shown in Table 5, Table 6, and Table 21, the 2M-parameter MiniX foundation model achieves strong classification and regression performance, outperforming TabPFN-v2 (7M), TabICL (27M), and Mitra (75M), and trailing only slightly behind LimiX (16M). These results underscore the effectiveness of RaBEL and the proposed Reordered Bidirectional Attention mechanism.

### 5.4 ABLATION STUDY

We conduct an ablation experiments to see the effectiveness of RaBEL and RBA(Reordered Bidirectional Attention mechanism), results displayed in Table 7. The Results directly validate the effectiveness of RaBEL and RBA.

Further, we perform an ablation study over the hyperparameters of RaBEL, sweeping settings such as the token embedding dimension, number of kernels, $\sigma$, initialization scheme, and kernel form. The results are reported in the Section A.4.

## 6 CONCLUSION

We presented **RaBEL**, a radial-basis embedding layer that replaces linear numeric encoders and resolves the early-layer low-rank bottleneck by providing localized, scale-aware representations. We also revisited bidirectional attention and demonstrated that it improves conditioning and guarantees that every attention computation contributes to prediction. The resulting model, **MiniX**, achieves superior accuracy with a substantially smaller parameter budget than prior tabular foundation models, while lowering compute. Future work will explore hybrid basis libraries, stronger self-supervised pretraining, and scaling MiniX to broader domains and distribution shifts.

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

## A APPENDIX

### A.1 PROOF OF THEORETICAL ANALYSES

**Proof of Proposition 3.1.** The embedding modules consist of two $p$-dimensional parameter vector $\alpha, \beta \in \mathbb{R}^{p \times 1}$. Thus $x_i$ is embedded as $z_i^T = x_i \alpha^T + \beta^T$, and the embedded input matrix $z^T = [z_1, z_2, ..., z_n]^T \in \mathbb{R}^{n \times p}$. The rank of the matrix is obviously at most 2, since all its row vectors can be written as linear combinations of $\alpha$ and $\beta$.

The parameter matrix of self-attention are: $W_Q, W_K, W_V \in \mathbb{R}^{p \times p}$. Thus the attention score of query $x_i$ to $x_j$ is:

$$
\begin{aligned}
a_{ij} &= (x_i \alpha^T + \beta^T) W_Q ((x_j \alpha^T + \beta^T) W_K)^T \\
&= x_i x_j \alpha^T W_Q W_K^T \alpha + x_i \alpha^T W_Q W_K^T \beta + x_j \beta^T W_Q W_K^T \alpha + \beta^T W_Q W_K^T \beta \\
&= \lambda_1 x_i x_j + \lambda_2 x_i + \lambda_3 x_j + \lambda_4
\end{aligned}
\tag{1}
$$

Where $\lambda_1 = \alpha^T W_Q W_K^T \alpha$, $\lambda_2 = \alpha^T W_Q W_K^T \beta$, $\lambda_3 = \beta^T W_Q W_K^T \alpha$, $\lambda_4 = \beta^T W_Q W_K^T \beta$. Thus we have:

$$
\begin{aligned}
\hat{z}_i^T &= \sum_{j=1}^{n} \frac{\exp(a_{ij}/\sqrt{p})}{\sum_{j=1}^{n} \exp(a_{ij}/\sqrt{p})} \cdot (x_j \alpha^T + \beta^T) W_V \\
&= \left( \sum_{j=1}^{n} \frac{x_j \exp(a_{ij}/\sqrt{p})}{\sum_{j=1}^{n} \exp(a_{ij}/\sqrt{p})} \right) \alpha_V^T + \beta_V^T
\end{aligned}
\tag{2}
$$

Where $\alpha_V^T = \alpha^T W_V$ and $\beta_V^T = \beta^T W_V$. We can see that $z_i$ can also be written as the linear combination of two vectors $\alpha_V$ and $\beta_V$, thus the rank of the output is at most 2. For multi-head attention, the calculation of each head is exactly the same as standard self-attention. Use $W_O^{(h)}$ to denote the output projection matrix of head $h$. The output projection is:

$$
\begin{aligned}
\hat{z}_i &= \sum_{h=1}^{H} \hat{z}_{i,h}^T W_O^{(h)} = \sum_{h=1}^{H} (\mu_{i,h} \alpha_V^{(h)} + \beta_V^{(h)})^T W_O^{(h)} \\
&= \sum_{h=1}^{H} (\mu_{i,h} \alpha_O^{(h)} + \beta_O^{(h)})^T \\
&= \sum_{h=1}^{H} (\mu_{i,h} \alpha_O^{(h)})^T + \beta_O^T
\end{aligned}
\tag{3}
$$

Thus $\hat{z}_i$ can be written as the linear combinations of $H + 1$ vectors, indicating that the rank of the output is at most $H + 1$.

## A.2 Low-Rank Approximation of TabPFN-v2

We conduct experiments of low-rank approximation for TabPFN-v2 on OpenML-CC18 to further verify the low-rank problem. Here each layer consists of three modules: feature attention, sample attention, a feed-forward network (FFN). Thus there are 36 modules for the 12-layer network. We also flatten the dimensions of samples and features to obtain a 2D matrix. Then we conduct SVD and set the singular values to zero except the largest $r$ ones and recover the matrix as the low-rank approximation. Note that we conduct such approximation as many as 36 times for each forward pass. From Table 8, we can see that the AUC is still very close to the full rank setting even it has reduced to 50, which is about 1/4 of the hidden dimension size. It still shows a competitive performance even when the rank has even reduced to 20, which is nearly 1/10 of the hidden dimension size. Such a phenomenon indicates that the current utilization of the hidden space is extremely inefficient, and emphasizes the need of developing a better embedding strategy.

Table 8: AUC of TabPFN-v2 after low-rank approximation. We can see that the model consistently shows a competive performance as the rank decreases to 20.

| Rank | 192 | 100 | 75 | 50 | 40 | 30 | 20 | 10 | 5 |
|---|---|---|---|---|---|---|---|---|---|
| AUC | 0.9177 | 0.9179 | 0.9175 | 0.9143 | 0.9100 | 0.9052 | 0.8985 | 0.8636 | 0.7674 |

## A.3 Toy Experiments of RBA

We validated SNF using a synthetic dataset derived from a Directed Acyclic Graph (DAG) to test its ability to capture feature dependencies. Figure 2 (a) shows the DAG (target $y$ shaded) alongside the feature-attention maps in Figure 2 (b) (c). In contrast to FSN, which relies heavily on self-attention, SNF demonstrates lower self-attention and effectively targets other features. Crucially, SNF assigns significantly higher attention scores to the direct causes of $y$ (e.g., Node 0 in row 1), confirming its superior performance in modeling feature interactions.

## A.4 Ablation on RaBEL

We conducted a comprehensive ablation study to investigate the impact of key hyperparameters and design choices on model performance (Tables 9–14). regarding model capacity, we observe

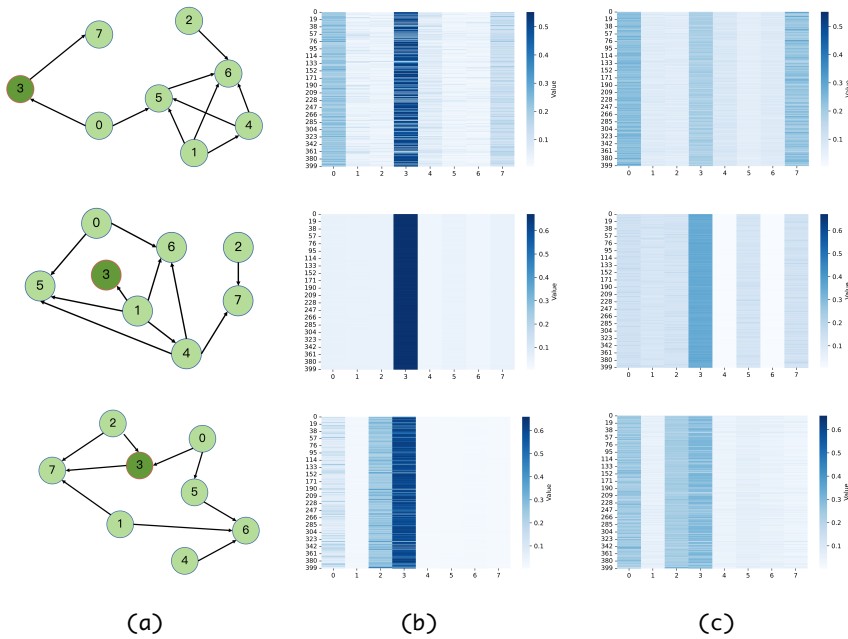

(a)          (b)          (c)

Figure 2: Visualization of attention scores (a) DAGs of the generated datasets. (b) Feature Attention Heatmap of FSN. (c) Feature Attention Heatmap of SNF. While FSN is dominated by self-attention, SNF demonstrates a broader attentional span that effectively targets neighboring features.

that an embedding dimension of 32 achieves the optimal trade-off (Table 9), while increasing the number of kernels consistently improves representation power, peaking at 64 kernels (Table 10). For kernel configuration, a fixed bandwidth $\sigma = 1.0$ with uniformly distributed kernels proves most effective, outperforming learnable $\sigma$ and random distributions (Table 11, 13, 14). Most notably, the initialization strategy plays a critical role; as shown in Table 12, orthogonal initialization yields a substantial performance gain (reaching 89.03% AUC) compared to standard Xavier Glorot & Bengio (2010) or Kaiming He et al. (2015) initializations, highlighting the importance of orthogonality in the exponent embedding.

Table 9: Impact of token embedding dimension

| dim | AUC | Acc. | F1 |
|---|---|---|---|
| 16 | 84.51 | 77.68 | 68.32 |
| 32 | **85.17** | **77.95** | **69.12** |
| 64 | 84.78 | 77.82 | 68.71 |

Table 10: Impact of number of kernels

| n kernels | AUC | Acc. | F1 |
|---|---|---|---|
| 16 | 84.20 | 77.12 | 68.53 |
| 32 | 84.76 | 77.68 | 68.88 |
| 64 | **85.19** | **77.95** | **69.08** |

Table 11: Impact of $\sigma$ value

| $\sigma$ | AUC | Acc. | F1 |
|---|---|---|---|
| 0.5 | 84.66 | 77.62 | 68.85 |
| 1.0 | **84.75** | **77.93** | **69.01** |
| 2.0 | 84.67 | 77.44 | 68.04 |

Table 12: Exponent embedding init method

| method | AUC | Acc. | F1 |
|---|---|---|---|
| orthogonal | **89.03** | **77.67** | **68.95** |
| xavier | 84.80 | 77.57 | 68.53 |
| kaiming | 84.68 | 77.44 | 68.27 |

Table 13: Impact of kernels form

| form | AUC | Acc. | F1 |
|---|---|---|---|
| random | 84.43 | 77.67 | 68.55 |
| uniform | **84.99** | **77.93** | **69.07** |

Table 14: Learnable vs. Fixed $\sigma$ (init $\sigma = 1$)

| mode | AUC | Acc. | F1 |
|---|---|---|---|
| learn | 84.65 | 77.59 | 68.53 |
| fix | **84.81** | **77.93** | **69.02** |

## A.5 RESULTS ON OPEN-SOURCE BENCHMARKS

Table 15: Classification results on **BCCO-CLS**, sorted by mean AUC in descending order, with the parameter counts of all foundation models highlighted in blue.

| | BCCO-CLS | | | | | |
|---|---|---|---|---|---|---|
| | Mean | | | Rank | | |
| Model | AUC (↑) | Acc. (↑) | F1 (↑) | AUC (↓) | Acc. (↓) | F1 (↓) |
| LimiX (16.52M) | 0.871 | 0.804 | 0.731 | 2.679 | 2.387 | 3.075 |
| MiniX (1.92M) | 0.858 | 0.787 | 0.701 | 6.406 | 6.689 | 8.830 |
| TabICL (27.10M) | 0.847 | 0.768 | 0.672 | 7.623 | 9.226 | 11.396 |
| AutoGluon | 0.846 | 0.771 | 0.677 | 8.792 | 8.943 | 10.425 |
| TabPFN-v2 (7.24M) | 0.843 | 0.772 | 0.679 | 9.575 | 10.274 | 11.896 |
| XGBoost | 0.834 | 0.762 | 0.674 | 11.160 | 11.491 | 12.217 |
| Mitra (75.67M) | 0.836 | 0.764 | 0.664 | 11.566 | 12.321 | 14.236 |
| LightGBM | 0.832 | 0.763 | 0.678 | 12.189 | 11.406 | 12.057 |
| TabM | 0.827 | 0.763 | 0.666 | 12.660 | 11.670 | 12.292 |
| RF | 0.829 | 0.756 | 0.652 | 12.802 | 13.104 | 14.453 |
| CatBoost | 0.829 | 0.757 | 0.664 | 13.358 | 12.472 | 13.264 |
| ET | 0.825 | 0.745 | 0.618 | 13.953 | 15.557 | 18.368 |
| RealMLP | 0.824 | 0.759 | 0.673 | 14.717 | 13.245 | 12.019 |
| FT-Transformer | 0.813 | 0.744 | 0.642 | 15.189 | 15.698 | 16.009 |
| T2G-Former | 0.808 | 0.742 | 0.646 | 16.255 | 16.009 | 15.151 |
| ModernNCA | 0.815 | 0.752 | 0.658 | 17.123 | 17.292 | 16.868 |
| MLP-PLR | 0.804 | 0.733 | 0.635 | 17.481 | 17.783 | 16.925 |
| ExcelFormer | 0.810 | 0.742 | 0.655 | 17.698 | 17.019 | 15.406 |
| TabR | 0.809 | 0.750 | 0.657 | 18.179 | 15.925 | 14.915 |
| DCN-v2 | 0.794 | 0.725 | 0.618 | 18.660 | 18.708 | 18.274 |
| ResNet | 0.800 | 0.728 | 0.641 | 18.717 | 18.981 | 17.594 |
| TANGOS | 0.799 | 0.731 | 0.641 | 19.038 | 18.689 | 17.028 |
| MLP | 0.787 | 0.720 | 0.614 | 20.349 | 19.774 | 20.575 |
| SAINT | 0.791 | 0.726 | 0.623 | 20.594 | 19.406 | 17.953 |
| AutoInt | 0.779 | 0.718 | 0.601 | 22.132 | 21.358 | 21.547 |
| DANets | 0.771 | 0.705 | 0.601 | 22.368 | 21.396 | 21.642 |
| SNN | 0.773 | 0.708 | 0.584 | 22.585 | 22.094 | 22.245 |
| TabTransformer | 0.762 | 0.699 | 0.566 | 22.594 | 21.764 | 22.151 |
| SwitchTab | 0.766 | 0.700 | 0.590 | 23.509 | 23.972 | 22.755 |
| TabCaps | 0.744 | 0.701 | 0.580 | 25.755 | 23.708 | 24.066 |
| NODE | 0.754 | 0.695 | 0.531 | 26.000 | 24.453 | 27.226 |
| TabNet | 0.712 | 0.685 | 0.561 | 29.217 | 26.557 | 26.613 |
| GrowNet | 0.682 | 0.641 | 0.522 | 29.679 | 29.358 | 28.302 |

## A.6 INFERENCE SPEED COMPARISON

We evaluate inference efficiency using a synthetic classification dataset comprising 900 samples and 60 features. All results are reported as the average of three runs on an AMD EPYC 9354 CPU and an NVIDIA RTX 4090 GPU (Table 24). MiniX achieves a remarkable inference time of 171.40 ms on GPU, outperforming TabPFN-v2 by $\approx 2\times$ and TabICL by $> 10\times$. Furthermore, MiniX maintains a substantial lead in CPU environments against heavy models like Mitra, confirming its practical deployability.

## A.7 RANK COMPARISON BETWEEN MINIX AND 2M BASELINE

To verify that the observed rank improvement stems from RaBEL rather than the experimental and optimization settings adopted from LimiX, we introduced a 2M-parameter baseline. This baseline shares the exact same training configuration and layer order of SNF as MiniX, with the sole difference being the embedding layer: the baseline uses a Linear projection, whereas MiniX employs

Table 16: Regression results on **BCCO-REG**, sorted by mean $R^2$ in descending order, with the parameter counts of all foundation models highlighted in blue.

| | BCCO-REG | | | |
|---|---|---|---|---|
| | Mean | | Rank | |
| Model | $R^2$ ($\uparrow$) | RMSE ($\downarrow$) | $R^2$ ($\downarrow$) | RMSE ($\downarrow$) |
| LimiX (16.52M) | 0.794 | 0.386 | 3.860 | 4.700 |
| AutoGluon | 0.781 | 0.398 | 5.140 | 5.120 |
| TabM | 0.773 | 0.397 | 5.400 | 5.400 |
| RealMLP | 0.766 | 0.402 | 5.960 | 5.880 |
| TabPFN-v2 (7.24M) | 0.772 | 0.404 | 6.440 | 6.340 |
| MiniX (1.92M) | 0.785 | 0.392 | 6.580 | 6.460 |
| XGBoost | 0.764 | 0.415 | 9.740 | 9.780 |
| LightGBM | 0.715 | 0.423 | 10.060 | 10.040 |
| ET | 0.757 | 0.431 | 12.180 | 12.120 |
| CatBoost | 0.741 | 0.427 | 12.660 | 12.460 |
| RF | 0.752 | 0.438 | 13.460 | 13.440 |
| ExcelFormer | 0.743 | 0.443 | 13.640 | 13.700 |
| T2G-Former | 0.743 | 0.442 | 14.940 | 14.880 |
| FT-Transformer | 0.737 | 0.448 | 15.520 | 15.500 |
| DCN-v2 | 0.739 | 0.448 | 15.840 | 15.840 |
| TabR | 0.733 | 0.448 | 16.580 | 16.660 |
| ResNet | 0.720 | 0.468 | 17.100 | 17.060 |
| ModernNCA | 0.598 | 0.471 | 17.300 | 17.200 |
| Mitra (75.67M) | 0.667 | 0.474 | 17.700 | 17.620 |
| TANGOS | 0.719 | 0.468 | 17.740 | 17.740 |
| MLP-PLR | 0.734 | 0.453 | 17.800 | 17.800 |
| SAINT | 0.701 | 0.481 | 18.540 | 18.560 |
| MLP | 0.701 | 0.487 | 19.020 | 19.080 |
| AutoInt | 0.724 | 0.465 | 19.180 | 19.100 |
| NODE | 0.643 | 0.543 | 22.100 | 22.020 |
| TabNet | 0.670 | 0.516 | 22.720 | 22.660 |
| DNNR | -2.152 | 1.329 | 22.860 | 22.880 |
| SNN | 0.434 | 0.720 | 27.220 | 27.220 |
| GrowNet | 0.201 | 0.864 | 28.880 | 28.860 |
| DANets | 0.005 | 0.979 | 30.480 | 30.500 |
| TabTransformer | -0.000 | 0.981 | 30.640 | 30.660 |
| SwitchTab | 0.001 | 0.981 | 30.720 | 30.720 |

Table 17: Classification results on **CC18**, sorted by mean AUC in descending order, with the parameter counts of all foundation models highlighted in blue.

| | OpenML-cc18 | | | | | |
| | Mean | | | Rank | | |
| Model | AUC (↑) | Acc. (↑) | F1 (↑) | AUC (↓) | Acc. (↓) | F1 (↓) |
|---|---|---|---|---|---|---|
| LimiX (16.52M) | 0.939 | 0.893 | 0.807 | 4.475 | 5.712 | 4.780 |
| MiniX (1.92M) | 0.935 | 0.892 | 0.799 | 5.475 | 5.695 | 6.322 |
| AutoGluon | 0.931 | 0.885 | 0.785 | 7.254 | 7.814 | 8.305 |
| TabPFN-v2 (7.24M) | 0.929 | 0.887 | 0.786 | 8.593 | 8.305 | 8.780 |
| TabICL (27.10M) | 0.933 | 0.881 | 0.786 | 8.915 | 10.695 | 10.508 |
| TabM | 0.926 | 0.881 | 0.775 | 9.339 | 11.559 | 11.339 |
| RealMLP | 0.925 | 0.883 | 0.789 | 10.712 | 9.153 | 8.220 |
| XGBoost | 0.928 | 0.879 | 0.770 | 10.746 | 11.864 | 11.932 |
| LightGBM | 0.927 | 0.879 | 0.770 | 11.102 | 11.102 | 11.271 |
| CatBoost | 0.926 | 0.870 | 0.765 | 13.322 | 14.729 | 14.746 |
| Mitra (75.67M) | 0.922 | 0.869 | 0.739 | 13.712 | 15.102 | 16.915 |
| TANGOS | 0.914 | 0.868 | 0.758 | 14.915 | 14.814 | 15.000 |
| ExcelFormer | 0.919 | 0.872 | 0.770 | 14.966 | 13.915 | 13.763 |
| RF | 0.925 | 0.872 | 0.757 | 15.034 | 14.305 | 15.339 |
| ET | 0.924 | 0.864 | 0.717 | 15.051 | 17.017 | 19.153 |
| ResNet | 0.917 | 0.865 | 0.765 | 15.102 | 14.983 | 14.068 |
| MLP | 0.913 | 0.861 | 0.742 | 15.508 | 15.712 | 16.797 |
| T2G-Former | 0.912 | 0.864 | 0.748 | 16.864 | 16.407 | 16.203 |
| MLP-PLR | 0.902 | 0.864 | 0.733 | 17.254 | 17.017 | 17.407 |
| ModernNCA | 0.912 | 0.865 | 0.747 | 18.153 | 17.475 | 16.797 |
| TabR | 0.907 | 0.870 | 0.757 | 18.254 | 14.797 | 14.441 |
| FT-Transformer | 0.909 | 0.862 | 0.737 | 18.373 | 16.864 | 17.983 |
| AutoInt | 0.909 | 0.852 | 0.727 | 19.864 | 19.847 | 19.898 |
| DANets | 0.904 | 0.844 | 0.712 | 19.898 | 19.288 | 20.475 |
| DCN-v2 | 0.904 | 0.856 | 0.727 | 20.525 | 19.339 | 19.966 |
| SNN | 0.901 | 0.839 | 0.694 | 22.441 | 22.186 | 22.847 |
| SwitchTab | 0.887 | 0.823 | 0.666 | 23.441 | 22.966 | 23.712 |
| TabCaps | 0.877 | 0.852 | 0.688 | 23.881 | 20.763 | 22.373 |
| TabTransformer | 0.854 | 0.799 | 0.631 | 24.492 | 23.475 | 24.186 |
| NODE | 0.894 | 0.813 | 0.622 | 25.237 | 26.305 | 28.390 |
| SAINT | 0.531 | 0.508 | 0.451 | 26.814 | 26.288 | 24.831 |
| TabNet | 0.869 | 0.822 | 0.664 | 27.390 | 25.034 | 26.780 |
| GrowNet | 0.819 | 0.749 | 0.571 | 27.983 | 28.271 | 27.559 |

Table 18: Classification results on **PFN-CLS**, sorted by mean AUC in descending order, with the parameter counts of all foundation models highlighted in blue.

| | PFN-CLS | | | | | |
| | Mean | | | Rank | | |
| Model | AUC (↑) | Acc. (↑) | F1 (↑) | AUC (↓) | Acc. (↓) | F1 (↓) |
|---|---|---|---|---|---|---|
| LimiX (16.52M) | 0.923 | 0.862 | 0.786 | 2.207 | 3.276 | 3.138 |
| MiniX (1.92M) | 0.913 | 0.848 | 0.766 | 4.207 | 6.828 | 6.103 |
| TabPFN-v2 (7.24M) | 0.910 | 0.845 | 0.756 | 5.828 | 7.586 | 7.690 |
| AutoGluon | 0.906 | 0.835 | 0.738 | 6.207 | 8.207 | 8.000 |
| TabICL (27.10M) | 0.903 | 0.832 | 0.742 | 7.414 | 9.241 | 9.724 |
| XGBoost | 0.898 | 0.831 | 0.733 | 9.655 | 8.069 | 9.103 |
| Mitra (75.67M) | 0.897 | 0.826 | 0.719 | 10.069 | 10.828 | 12.655 |
| LightGBM | 0.893 | 0.826 | 0.725 | 10.103 | 9.483 | 10.034 |
| TabM | 0.895 | 0.829 | 0.734 | 10.690 | 10.276 | 10.724 |
| CatBoost | 0.895 | 0.819 | 0.720 | 11.276 | 11.034 | 12.379 |
| RealMLP | 0.889 | 0.823 | 0.732 | 11.655 | 12.897 | 12.448 |
| RF | 0.896 | 0.822 | 0.721 | 12.172 | 11.724 | 12.897 |
| ET | 0.893 | 0.809 | 0.675 | 13.103 | 14.483 | 16.379 |
| ExcelFormer | 0.883 | 0.812 | 0.713 | 15.655 | 13.379 | 14.828 |
| ResNet | 0.869 | 0.801 | 0.700 | 16.414 | 16.586 | 16.000 |
| TANGOS | 0.865 | 0.797 | 0.698 | 17.310 | 18.000 | 16.310 |
| MLP | 0.866 | 0.795 | 0.695 | 18.586 | 17.690 | 17.207 |
| T2G-Former | 0.848 | 0.792 | 0.688 | 19.621 | 18.034 | 17.931 |
| ModernNCA | 0.860 | 0.798 | 0.692 | 19.724 | 17.828 | 16.655 |
| FT-Transformer | 0.849 | 0.789 | 0.683 | 20.414 | 19.931 | 18.621 |
| SwitchTab | 0.858 | 0.776 | 0.626 | 20.586 | 21.517 | 21.103 |
| MLP-PLR | 0.848 | 0.786 | 0.650 | 20.724 | 21.034 | 22.103 |
| DANets | 0.844 | 0.770 | 0.631 | 21.138 | 20.724 | 21.862 |
| TabCaps | 0.834 | 0.788 | 0.636 | 22.448 | 20.276 | 21.448 |
| TabR | 0.842 | 0.789 | 0.688 | 23.034 | 19.655 | 18.241 |
| TabTransformer | 0.821 | 0.761 | 0.604 | 23.517 | 22.621 | 24.414 |
| DCN-v2 | 0.846 | 0.771 | 0.633 | 24.172 | 24.862 | 25.172 |
| NODE | 0.844 | 0.754 | 0.535 | 24.241 | 25.828 | 28.034 |
| AutoInt | 0.838 | 0.772 | 0.640 | 24.345 | 23.172 | 23.069 |
| SAINT | 0.708 | 0.669 | 0.563 | 24.448 | 23.379 | 22.310 |
| SNN | 0.831 | 0.762 | 0.595 | 25.517 | 23.069 | 25.103 |
| TabNet | 0.825 | 0.768 | 0.631 | 27.207 | 24.897 | 26.069 |
| GrowNet | 0.756 | 0.689 | 0.497 | 29.862 | 28.655 | 28.172 |

Table 19: Regression results on **PFN-REG**, sorted by mean $R^2$ in descending order, with the parameter counts of all foundation models highlighted in blue.

| | PFN-REG | | | |
| | Mean | | Rank | |
| Model | $R^2$ ($\uparrow$) | RMSE ($\downarrow$) | $R^2$ ($\downarrow$) | RMSE ($\downarrow$) |
|---|---|---|---|---|
| LimiX (16.52M) | 0.682 | 0.468 | 5.148 | 6.148 |
| MiniX (1.92M) | 0.687 | 0.464 | 5.889 | 5.889 |
| RealMLP | 0.675 | 0.471 | 7.519 | 7.407 |
| AutoGluon | 0.669 | 0.484 | 7.593 | 7.593 |
| TabM | 0.677 | 0.468 | 7.667 | 7.556 |
| TabPFN-v2 (7.24M) | 0.667 | 0.478 | 8.630 | 8.370 |
| XGBoost | 0.661 | 0.493 | 10.481 | 10.481 |
| LightGBM | 0.656 | 0.499 | 11.111 | 11.037 |
| CatBoost | 0.652 | 0.501 | 12.519 | 12.444 |
| ET | 0.643 | 0.520 | 12.630 | 12.556 |
| Mitra (75.67M) | 0.620 | 0.534 | 14.333 | 14.407 |
| RF | 0.634 | 0.529 | 14.889 | 14.963 |
| T2G-Former | 0.630 | 0.507 | 15.037 | 14.926 |
| ExcelFormer | 0.637 | 0.517 | 15.407 | 15.444 |
| FT-Transformer | 0.627 | 0.511 | 15.741 | 15.630 |
| MLP | 0.565 | 0.582 | 15.926 | 16.074 |
| MLP-PLR | 0.629 | 0.516 | 16.074 | 16.148 |
| TabR | 0.628 | 0.511 | 16.148 | 16.000 |
| ModernNCA | 0.636 | 0.512 | 16.222 | 16.407 |
| DCN-v2 | 0.629 | 0.514 | 16.444 | 16.259 |
| ResNet | 0.588 | 0.559 | 16.481 | 16.519 |
| TANGOS | 0.576 | 0.568 | 16.926 | 17.074 |
| SAINT | -8.420 | 0.618 | 17.926 | 17.519 |
| AutoInt | 0.599 | 0.544 | 20.185 | 20.074 |
| NODE | 0.487 | 0.668 | 21.741 | 21.556 |
| TabNet | 0.416 | 0.647 | 22.963 | 22.889 |
| DNNR | -8.173 | 2.289 | 24.222 | 24.370 |
| SNN | 0.364 | 0.764 | 26.185 | 26.222 |
| GrowNet | 0.087 | 0.939 | 27.556 | 27.519 |
| DANets | 0.001 | 0.988 | 28.815 | 28.852 |
| SwitchTab | -0.025 | 1.001 | 29.259 | 29.296 |
| TabTransformer | -0.020 | 0.998 | 30.333 | 30.370 |

Table 20: Regression results on **TabArena-REG**, sorted by mean $R^2$ in descending order, with the parameter counts of all foundation models highlighted in blue.

| | TabArena-REG | | | |
| | Mean | | Rank | |
| Model | $R^2$ ($\uparrow$) | RMSE ($\downarrow$) | $R^2$ ($\downarrow$) | RMSE ($\downarrow$) |
|---|---|---|---|---|
| AutoGluon | 0.791 | 0.414 | 3.462 | 3.462 |
| LimiX (16.52M) | 0.796 | 0.406 | 3.462 | 3.462 |
| MiniX (1.92M) | 0.788 | 0.413 | 4.538 | 4.538 |
| TabPFN-v2 (7.24M) | 0.777 | 0.422 | 5.923 | 5.923 |
| TabM | 0.777 | 0.424 | 6.692 | 6.692 |
| CatBoost | 0.774 | 0.431 | 7.308 | 7.308 |
| XGBoost | 0.778 | 0.430 | 7.462 | 7.462 |
| RealMLP | 0.776 | 0.426 | 8.000 | 8.000 |
| LightGBM | 0.771 | 0.435 | 8.000 | 8.000 |
| RF | 0.758 | 0.456 | 11.000 | 11.000 |
| ET | 0.746 | 0.464 | 11.385 | 11.308 |
| TabR | 0.729 | 0.476 | 15.231 | 15.231 |
| ExcelFormer | 0.681 | 0.521 | 15.385 | 15.385 |
| NODE | 0.665 | 0.542 | 15.692 | 15.692 |
| TANGOS | 0.673 | 0.534 | 16.538 | 16.538 |
| DCN-v2 | 0.718 | 0.486 | 16.692 | 16.769 |
| Mitra (75.67M) | 0.666 | 0.538 | 17.692 | 17.692 |
| ModernNCA | 0.712 | 0.496 | 18.462 | 18.462 |
| MLP-PLR | 0.714 | 0.496 | 18.538 | 18.538 |
| AutoInt | 0.705 | 0.503 | 19.154 | 19.154 |
| MLP | 0.694 | 0.516 | 19.615 | 19.615 |
| ResNet | 0.687 | 0.521 | 19.769 | 19.769 |
| SAINT | 0.712 | 0.498 | 20.250 | 20.250 |
| T2G-Former | 0.677 | 0.526 | 20.538 | 20.538 |
| TabNet | 0.641 | 0.564 | 20.769 | 20.769 |
| FT-Transformer | 0.626 | 0.572 | 23.077 | 23.077 |
| DNNR | -0.368 | 1.058 | 25.615 | 25.615 |
| SNN | 0.451 | 0.729 | 26.692 | 26.692 |
| GrowNet | 0.316 | 0.814 | 28.385 | 28.385 |
| TabTransformer | 0.016 | 0.993 | 30.333 | 30.333 |
| DANets | 0.012 | 0.998 | 30.462 | 30.462 |
| SwitchTab | 0.004 | 1.002 | 30.923 | 30.923 |

Table 21: Classification results on **TabZilla**.

| Model | TabZilla | | | | | |
|---|---|---|---|---|---|---|
| | Mean | | | Rank | | |
| | AUC (↑) | Acc. (↑) | F1 (↑) | AUC (↓) | Acc. (↓) | F1 (↓) |
| LimiX (16.52M) | 0.943 | 0.885 | 0.836 | 5.037 | 5.333 | 6.519 |
| MiniX (1.92M) | 0.938 | 0.883 | 0.832 | 5.963 | 6.704 | 7.074 |
| AutoGluon | 0.933 | 0.871 | 0.803 | 7.889 | 8.259 | 9.704 |
| TabPFN-v2 (7.24M) | 0.929 | 0.863 | 0.797 | 8.704 | 9.815 | 10.185 |
| TabICL (27.10M) | 0.933 | 0.864 | 0.803 | 9.704 | 10.556 | 11.444 |
| XGBoost | 0.929 | 0.863 | 0.789 | 10.111 | 11.407 | 12.778 |
| TabM | 0.928 | 0.869 | 0.816 | 10.148 | 9.889 | 10.407 |
| LightGBM | 0.927 | 0.863 | 0.796 | 11.963 | 11.074 | 11.556 |
| RF | 0.924 | 0.852 | 0.773 | 12.444 | 13.519 | 14.037 |
| RealMLP | 0.923 | 0.872 | 0.815 | 12.481 | 9.185 | 9.333 |
| CatBoost | 0.922 | 0.848 | 0.780 | 13.778 | 14.889 | 14.852 |
| Mitra (75.67M) | 0.915 | 0.841 | 0.758 | 15.815 | 15.667 | 16.593 |
| ExcelFormer | 0.915 | 0.861 | 0.802 | 15.852 | 13.481 | 13.481 |
| T2G-Former | 0.909 | 0.852 | 0.790 | 15.926 | 16.000 | 16.222 |
| ModernNCA | 0.907 | 0.850 | 0.794 | 16.000 | 16.593 | 15.778 |
| ET | 0.912 | 0.837 | 0.745 | 16.370 | 16.704 | 18.000 |
| TabR | 0.904 | 0.853 | 0.793 | 16.852 | 15.074 | 15.481 |
| MLP-PLR | 0.906 | 0.847 | 0.773 | 16.889 | 16.519 | 17.630 |
| TANGOS | 0.909 | 0.841 | 0.776 | 17.000 | 15.519 | 15.963 |
| FT-Transformer | 0.903 | 0.842 | 0.769 | 17.778 | 17.593 | 17.111 |
| MLP | 0.903 | 0.825 | 0.747 | 18.111 | 18.407 | 18.889 |
| ResNet | 0.908 | 0.834 | 0.769 | 18.407 | 17.037 | 16.519 |
| AutoInt | 0.896 | 0.833 | 0.748 | 18.778 | 17.963 | 18.407 |
| DCN-v2 | 0.904 | 0.844 | 0.781 | 19.481 | 18.407 | 18.889 |
| SAINT | 0.824 | 0.764 | 0.680 | 21.037 | 20.778 | 20.444 |
| SNN | 0.874 | 0.816 | 0.706 | 22.519 | 20.556 | 21.778 |
| DANets | 0.881 | 0.800 | 0.712 | 22.926 | 22.000 | 22.519 |
| TabTransformer | 0.814 | 0.759 | 0.659 | 23.296 | 21.630 | 22.593 |
| SwitchTab | 0.860 | 0.764 | 0.660 | 23.741 | 25.519 | 25.111 |
| TabCaps | 0.887 | 0.816 | 0.729 | 25.296 | 22.481 | 22.815 |
| NODE | 0.869 | 0.784 | 0.633 | 25.593 | 25.593 | 27.630 |
| GrowNet | 0.829 | 0.732 | 0.651 | 27.630 | 27.222 | 26.148 |
| TabNet | 0.860 | 0.771 | 0.668 | 28.593 | 28.000 | 27.556 |

Table 22: Classification results on **TALENT-CLS**, sorted by mean AUC in descending order, with the parameter counts of all foundation models highlighted in blue.

| | TALENT-CLS | | | | | |
| | Mean | | | Rank | | |
| Model | AUC (↑) | Acc. (↑) | F1 (↑) | AUC (↓) | Acc. (↓) | F1 (↓) |
|---|---|---|---|---|---|---|
| LimiX (16.52M) | 0.903 | 0.861 | 0.752 | 4.212 | 3.380 | 4.464 |
| MiniX (1.92M) | 0.897 | 0.853 | 0.734 | 6.067 | 6.006 | 7.525 |
| TabICL (27.10M) | 0.894 | 0.845 | 0.715 | 6.531 | 7.620 | 9.162 |
| TabPFN-v2 (7.24M) | 0.895 | 0.850 | 0.727 | 7.017 | 6.933 | 8.872 |
| AutoGluon | 0.891 | 0.845 | 0.719 | 7.285 | 7.911 | 8.732 |
| XGBoost | 0.881 | 0.837 | 0.713 | 10.782 | 10.955 | 11.285 |
| TabM | 0.881 | 0.842 | 0.719 | 10.961 | 10.486 | 10.257 |
| LightGBM | 0.880 | 0.836 | 0.713 | 11.117 | 11.302 | 11.609 |
| RealMLP | 0.881 | 0.843 | 0.726 | 11.749 | 9.693 | 9.067 |
| Mitra (75.67M) | 0.882 | 0.834 | 0.689 | 11.899 | 12.743 | 14.810 |
| CatBoost | 0.876 | 0.828 | 0.704 | 13.184 | 13.279 | 13.475 |
| RF | 0.877 | 0.828 | 0.691 | 14.101 | 14.676 | 15.765 |
| ET | 0.875 | 0.821 | 0.662 | 14.916 | 17.380 | 18.944 |
| ExcelFormer | 0.870 | 0.826 | 0.699 | 15.441 | 16.128 | 14.911 |
| ResNet | 0.866 | 0.825 | 0.695 | 16.944 | 16.693 | 14.994 |
| FT-Transformer | 0.859 | 0.822 | 0.678 | 17.771 | 17.939 | 17.542 |
| T2G-Former | 0.858 | 0.823 | 0.683 | 17.877 | 17.559 | 17.056 |
| TANGOS | 0.861 | 0.818 | 0.684 | 18.123 | 18.039 | 16.637 |
| MLP | 0.862 | 0.817 | 0.675 | 18.514 | 18.156 | 18.240 |
| ModernNCA | 0.861 | 0.825 | 0.683 | 19.112 | 18.732 | 17.972 |
| TabR | 0.858 | 0.824 | 0.680 | 19.385 | 17.866 | 16.978 |
| DCN-v2 | 0.854 | 0.815 | 0.662 | 19.894 | 19.978 | 19.715 |
| MLP-PLR | 0.849 | 0.816 | 0.663 | 20.603 | 19.223 | 19.179 |
| DANets | 0.848 | 0.805 | 0.654 | 21.251 | 21.000 | 20.559 |
| SAINT | 0.813 | 0.781 | 0.630 | 22.385 | 21.413 | 20.816 |
| AutoInt | 0.842 | 0.803 | 0.646 | 22.743 | 23.274 | 22.620 |
| SwitchTab | 0.842 | 0.795 | 0.637 | 23.480 | 23.972 | 23.123 |
| TabCaps | 0.834 | 0.813 | 0.654 | 23.536 | 20.313 | 20.737 |
| TabTransformer | 0.832 | 0.790 | 0.627 | 23.827 | 23.726 | 22.821 |
| SNN | 0.836 | 0.796 | 0.625 | 24.162 | 23.972 | 24.011 |
| NODE | 0.830 | 0.779 | 0.570 | 25.022 | 25.296 | 26.754 |
| TabNet | 0.818 | 0.794 | 0.630 | 27.285 | 24.765 | 25.173 |
| GrowNet | 0.743 | 0.704 | 0.542 | 29.553 | 28.978 | 26.905 |

Table 23: Regression results on **TALENT-REG**, sorted by mean $R^2$ in descending order, with the parameter counts of all foundation models highlighted in blue.

| | TALENT-REG | | | |
|---|---|---|---|---|
| | Mean | | Rank | |
| Model | $R^2$ ($\uparrow$) | RMSE ($\downarrow$) | $R^2$ ($\downarrow$) | RMSE ($\downarrow$) |
| LimiX (16.52M) | 0.735 | 0.433 | 3.232 | 3.919 |
| AutoGluon | 0.722 | 0.448 | 5.667 | 5.889 |
| TabM | 0.708 | 0.459 | 6.525 | 6.364 |
| TabPFN-v2 (7.24M) | 0.695 | 0.465 | 7.061 | 6.980 |
| MiniX (1.92M) | 0.721 | 0.451 | 7.061 | 6.980 |
| RealMLP | 0.697 | 0.465 | 7.202 | 7.030 |
| XGBoost | 0.710 | 0.462 | 8.434 | 8.384 |
| LightGBM | 0.707 | 0.461 | 9.283 | 9.253 |
| ET | 0.696 | 0.476 | 10.990 | 10.939 |
| CatBoost | 0.700 | 0.471 | 11.525 | 11.465 |
| RF | 0.697 | 0.474 | 11.596 | 11.596 |
| ExcelFormer | 0.653 | 0.512 | 14.687 | 14.727 |
| T2G-Former | 0.656 | 0.512 | 15.384 | 15.323 |
| TabR | 0.651 | 0.516 | 16.465 | 16.434 |
| DCN-v2 | -0.361 | 0.818 | 16.677 | 16.727 |
| Mitra (75.67M) | 0.602 | 0.547 | 16.889 | 16.909 |
| FT-Transformer | 0.648 | 0.519 | 16.960 | 16.909 |
| MLP-PLR | 0.653 | 0.521 | 17.222 | 17.182 |
| ModernNCA | 0.633 | 0.530 | 17.747 | 17.727 |
| ResNet | 0.562 | 0.550 | 17.879 | 17.848 |
| TANGOS | 0.592 | 0.547 | 18.384 | 18.364 |
| MLP | 0.556 | 0.564 | 18.768 | 18.808 |
| SAINT | -1.541 | 0.571 | 19.515 | 19.384 |
| AutoInt | 0.636 | 0.538 | 20.172 | 20.182 |
| NODE | 0.568 | 0.600 | 21.697 | 21.545 |
| TabNet | 0.576 | 0.586 | 22.869 | 22.818 |
| DNNR | -9.172 | 2.528 | 24.525 | 24.566 |
| SNN | 0.344 | 0.777 | 26.505 | 26.525 |
| GrowNet | -0.182 | 0.920 | 28.121 | 28.152 |
| DANets | 0.005 | 0.998 | 29.434 | 29.455 |
| TabTransformer | 0.001 | 1.001 | 29.576 | 29.626 |
| SwitchTab | -0.002 | 1.002 | 29.939 | 29.980 |

Table 24: Inference time comparison in milliseconds (ms). The experiments were conducted using an **AMD EPYC 9354 32-Core Processor** for CPU evaluation and an **NVIDIA GeForce RTX 4090** for GPU evaluation. The best results are highlighted in **bold**.

| Model | CPU (ms) | GPU (ms) |
|---|---|---|
| TabPFN-v2 | 51950.08 | 352.60 |
| LimiX | 68447.99 | 368.08 |
| TabICL | 22161.85 | 1749.61 |
| Mitra | 124453.05 | 5766.25 |
| MiniX | **17257.34** | **171.40** |

RaBEL. We report the rank comparison of the first three layers in the table, where the results demonstrate that MiniX yields a significant rank boost in the shallow layers.

Table 25: Comparison of Ranks. Subscripts denote improvement percentages.

| Model | Numerical Rank | Rank@99% | Rank@95% |
|---|---|---|---|
| Baseline-2M | 58.41 | 13.94 | 6.73 |
| MiniX | **78.62**(+34.60%) | **25.35**(+81.98%) | **12.31**(+83.18%) |

## A.8 FINE-GRAINED DATASET-LEVEL COMPARISON

We conducted a fine-grained dataset-level comparison on the TabArena (classification) and CTR23 (regression) benchmarks to evaluate the number of datasets where each model achieves the leading performance. Our comparison set includes established baselines such as TabPFN-v2, TabICL, Mitra, XGBoost, and CatBoost.

Notably, we exclude LimiX from this specific visualization. Since LimiX achieves state-of-the-art results on the vast majority of datasets, including it would obscure the relative performance dynamics between MiniX and the other baselines. We compare both our Baseline-2M and MiniX against this subset of models, as shown in Figures 3 to 6.

The results indicate that the Baseline-2M—trained using the LimiX paradigm—achieves performance comparable to XGBoost but slightly trails behind TabICL and TabPFN-v2. In contrast, MiniX consistently surpasses these baselines. Additionally, Mitra is omitted from the plot as it secured the leading position on a negligible number of datasets.

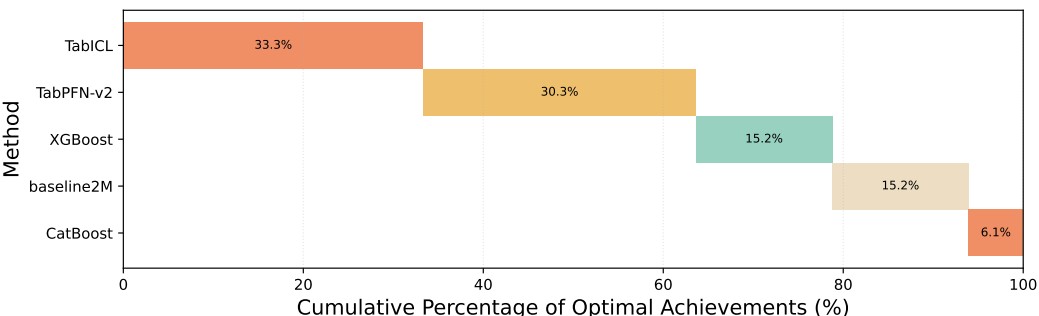

Figure 3: Cumulative Percentage of Optimal Achievements of Baseline2M on TabArena.

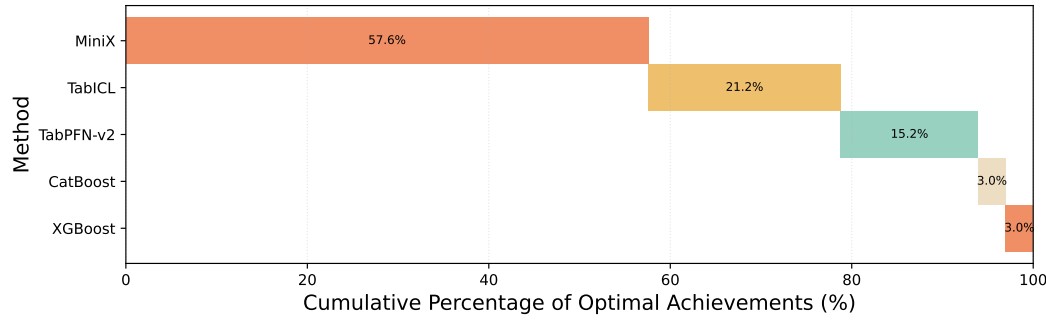

Figure 4: Cumulative Percentage of Optimal Achievements of MiniX on TabArena.

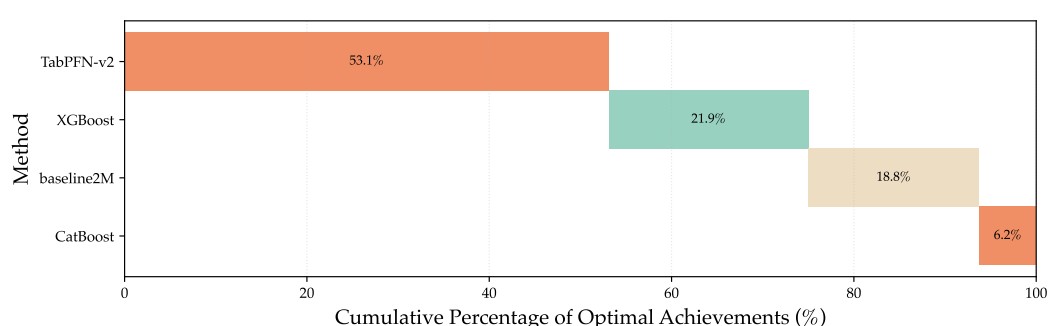

Figure 5: Cumulative Percentage of Optimal Achievements of Baseline2M on CTR23.

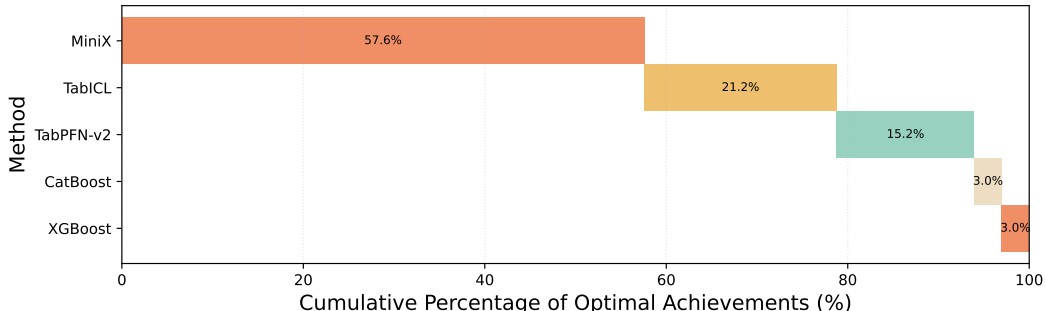

Figure 6: Cumulative Percentage of Optimal Achievements of MiniX on CTR23.

