# OpenReview forum: "RaBEL: Scale-Aware Radial-Basis Embeddings for Tabular Foundation Models"
_ICLR.cc/2026/Conference — Submitted to ICLR 2026_

### Official Review · Reviewer_hNSy · 2025-10-24

**Soundness:** 2
**Presentation:** 2
**Contribution:** 3
**Rating:** 2
**Confidence:** 4

**Summary:**

The paper proposes a new embedding scheme for tabular foundation models, such as TabPFN, that addresses the low-rank problem encountered with linear embeddings. Moreover, the authors change the order of blocks in transformer layers, which improves the model. All these improvements result in MiniX.

**Strengths:**

Overall, the idea of improving tabular foundation models and making them smaller yet effective is a great outcome of the research.

1. I think the low-rank problem is crucial, and this finding is the most valuable result of the paper.
2. Figure 1 is also valuable.
3. A small tabular foundation model with comparable performance is a strong result.
4. The ablation study in Table 4 is important for the paper.

**Weaknesses:**

See the questions for details.

Weaknesses summary:
1. Section 4.2 is poorly written.
2. The toy experiments are currently misleading.
3. There is a lack of details about the experimental setup.
4. I think the title can be improved.
5. There is a lack of important ablations on embedding.

**Questions:**

I will divide my questions into two parts: technical and stylistic/paper-related. You can find a summary at the end.

**Technical:**

1. I believe TabPFNv2 layer has the order `feats_attn -> sample_attn -> ffn`. Do you retain the final FFN layer, resulting in `sample_attn -> ffn -> feats_attn -> ffn`? If so, can you confirm that the improvement comes from the reordering rather than from adding an additional FFN? This should be clarified in the main text.

2. L231-238 contain the statement "RaBEL increases $r_{eff}$ in shallow layers by providing diverse..." without any experimental evidence. Either include supporting artifacts (tables/figures) or remove this paragraph, please.

3. What is a "bounded set of exponent bins"? What are the specific values for $b_{min}$ and $b_{max}$? From my understanding, this represents a method to partition inputs into bins like $[2^{b}, 2^{b + 1})$. I believe this explanation could be made clearer.

4. **Toy experiments:** Periodic embedding actually can approximate the functions shown in Figure 2b, though the results heavily depend on the initialization of $c_i$ (which I view as a significant limitation of periodic embedding and would emphasize). The simplest demonstration involves initializing $c_i$ with $N(0, 20^2)$ for equation (5) and increasing $k = 64$. I can provide code if needed. This section must include experimental details—currently, there's no information about the setup whatsoever (learning rate, number of epochs, num train points, etc.).

5. RaBEL is compared against periodic embeddings in toy examples, but there's no comparison including MLP-RaBEL or MiniX-PLR, making this toy example comparison appear redundant. While I understand that training MiniX-PLR is costly, I believe `MLP-RaBEL vs MLP-PLR` would provide more valuable insights than the current toy examples section. Additionally, could periodic embedding potentially resolve the rank problem? Overall, in this and previous question I mean that the question "why RaBEL is better than other embeddings?" is not answered clearly.

6. **Main experiments:** As I understand, you completely followed LimiX. However, there's currently no information about the experimental setup. What normalization was applied across all datasets? How was HPO performed? How were ranks calculated? How do you establish statistical significance for the reported results? Why do the TabArena results diverge from the official benchmark (e.g., RealMLP performs significantly worse in your setup)? Is there any reason TabM was not included?

7. Are all RaBEL embedding components necessary? Why not simply make $c$ and $\sigma$ trainable and omit the exponent-gating and shared-gates? Ablations using MLP could help address this question.

8. (minor) The paper would benefit significantly from experiments on large datasets. Due to the lower hidden dimension, you can evaluate MiniX on substantial datasets.

9. Could you provide intuition on why flattening the sample+feature dimension is preferable to flattening the feature+D dimension?


**Paper-related:**

1. Section 4.2 requires substantial revision: it contains duplicate text, redundant discussion of $r_{eff}$, lacks hyperparameter values ($M, b_{min}, b_{max}, \tau, h_s$), and presents unclear motivation. Motivations (ii) and (iii) on lines 293-294 remain unexplained. The third motivation should be validated experimentally. I recommend the authors invest time in rewriting this section more clearly. Consider starting with the initial RBF kernel, then introducing enhancements sequentially (without text duplication) and experimentally prove their significance. Alternatively, retain only the final "full" version of the embedding explanation.

2. **Minor:** Please define "energy" on line 163. I assume full energy represents the sum of squared singular values.

3. **Title suggest:** I find the title and abstract potentially misleading (w.r.t. paper conent) and potentially harmful to the paper. The title suggests the embedding as the primary contribution, but I believe the paper mostly about MiniX and making tabular foundation models better by fixing low-rank problem. I recommend revising the title to something like (this is just a rough suggestion) "MiniX: Fixing the low-rank problem in tabular foundation models with RBF-based embedding" to better reflect the paper's results and impact on the field.

4. **Note for multiple sections:** Please include experimental setup and implementation details throughout.

Overall, I believe these findings hold significant value for the community and the field, but the current paper quality is low. My current score is 2, though I'm willing to increase it toward acceptance if most technical questions are addressed during the rebuttal. Questions 4-6 are particularly crucial for raising my score to 6.

---

> ### Author Response · Authors · 2025-11-26
>
> # Q-T 1: Clarification on RBA structure.
> We clarify that RBA does **not** introduce an additional FFN layer. The design is a strict reordering of the existing components within a block, ensuring the parameter count remains identical to the baseline. Specifically:
>
> - **TabPFN-v2 (Original):** Feature Attn $\rightarrow$ Sample Attn $\rightarrow$ FFN.
> - **MiniX (RBA):** Sample Attn $\rightarrow$ FFN $\rightarrow$ Feature Attn.
>
> We do not retain a final FFN after the Feature Attention in the RBA block. We have revised the main text to explicitly rule out the `Sample Attn -> FFN -> Feature Attn -> FFN` interpretation and clearly state that no new modules were added.
>
> # Q-T 2: Unsupported Claims in L231-238
> We agree with the reviewer that this statement lacked direct experimental support. Upon re-evaluation, we found this paragraph to be redundant to the main narrative.  We have removed Lines 231–238 in the revised manuscript.
>
> # Q-T 3: Clarification on "Bounded Set of Exponent Bins"
> Your understanding is largely correct: this method partitions inputs based on their magnitude, but in a **soft, differentiable manner** rather than hard binning. 1.  **Definition:** The set $\mathcal{B} = \{b_{\min}, \dots, b_{\max}\}$ consists of consecutive integers representing the **exponents** (orders of magnitude) relative to the base $\beta$ (where $\beta=2$ in our default setting). For example, a bin index $b=3$ corresponds to an input magnitude of approximately $2^3 = 8$. 2.  **Specific Values:** In our experiments, we set the range to $b_{\min} = [-16]$ and $b_{\max} = [16]$. These boundaries were chosen to cover the effective dynamic range of the standardized input features. 3.  **Mechanism:** Instead of assigning an input $x$ to a single bin (hard partition), we calculate a **soft assignment distribution** $\pi(b)$ over these bins using a Gaussian kernel on the log-magnitude $\log_\beta(|x|)$. This allows the network to smoothly interpolate scale information and remain fully differentiable end-to-end.

---

> ### Author Response · Authors · 2025-11-26
>
> # Q-T 4 & Q-T 5: Weak toy experiments and need for comparisons between RaBEL and other numerical embedding methods
>
> We sincerely appreciate the reviewer's feedback. Initially, we designed the toy experiments to provide an **intuitive visualization** of how different embedding methods impact the representation space. However, we agree that qualitative visualizations alone are insufficient and that rigorous quantitative benchmarks provide stronger evidence for the method's effectiveness.
>
> To address this, we have significantly expanded our evaluation to include comprehensive quantitative experiments.
>
> 1. Additional MLP-based Experiments
>
> We have supplemented the paper with extensive MLP-based experimental results. These updates have been incorporated into the main text in Lines 290–323. As shown in the table below, our method (RaBEL) consistently outperforms other baselines across the majority of datasets.
>
> **Table 1: Performance comparison on various datasets using MLP backbone.**
>
> | **Metric1**       | **GC (↑)** | **CP (↑)** | **AC (↑)** | **CB (↑)** | **UK (↑)** | **BN (↑)** | **MA (↑)** | **CD (↑)** | **MH (↑)** | **CC (↑)** | **HS (↑)** | **SC (↑)** | **MV (↑)** |
> | ---------------- | ---------- | ---------- | ---------- | ---------- | ---------- | ---------- | ---------- | ---------- | ---------- | ---------- | ---------- | ---------- | ---------- |
> | **MLP**          | 0.7537     | 0.8092     | 0.6560     | 0.8319     | 0.9850     | 0.7520     | **0.8896** | 0.4476     | 0.7811     | 0.6158     | 0.7497     | 0.1799     | **0.9999** |
> | **MLP-MLP**      | 0.8984     | 0.8496     | 0.8398     | 0.8924     | **0.9863** | 0.7518     | 0.8577     | 0.4768     | 0.8618     | 0.8947     | 0.8375     | 0.1749     | 0.9990     |
> | **MLP-PLE**      | 0.7393     | 0.8031     | 0.8262     | 0.8817     | 0.8416     | 0.7389     | 0.8609     | 0.2969     | 0.8281     | 0.8576     | 0.8279     | **0.1815** | 0.9730     |
> | **MLP-Periodic** | 0.7817     | 0.8895     | 0.8054     | 0.8926     | 0.9821     | **0.7586** | 0.8625     | 0.4095     | 0.8306     | **0.9068** | 0.8286     | 0.1786     | 0.9994     |
> | **MLP-RaBEL**    | **0.9831** | **0.9582** | **0.9061** | **0.8979** | 0.9677     | 0.7580     | 0.8789     | **0.5124** | **0.8818** | 0.8998     | **0.8401** | 0.1813     | 0.9995     |
>
>
> | **Metric2**       | **GC (↑)** | **CP (↑)** | **AC (↑)** | **CB (↑)** | **UK (↑)** | **BN (↑)** | **MA (↑)** | **CD (↓)** | **MH (↓)** | **CC (↓)** | **HS (↓)** | **SC (↓)** | **MV (↓)** |
> | ---------------- | ---------- | ---------- | ---------- | ---------- | ---------- | ---------- | ---------- | ---------- | ---------- | ---------- | ---------- | ---------- | ---------- |
> | **MLP**          | 0.3483     | 0.6756     | 0.6750     | 0.9677     | **0.9008** | 0.5757     | 0.9611     | 0.7218     | 0.4862     | 0.6072     | 0.5259     | 0.9051     | **0.0119** |
> | **MLP-MLP**      | 0.5393     | 0.7224     | 0.7870     | **0.9692** | **0.9008** | 0.5766     | 0.9657     | 0.7025     | 0.3863     | 0.3178     | 0.4238     | 0.9078     | 0.0321     |
> | **MLP-PLE**      | 0.2584     | 0.6890     | 0.7750     | 0.9677     | 0.5868     | 0.5596     | 0.9654     | 0.8143     | 0.4308     | 0.3697     | 0.4361     | **0.9042** | 0.1642     |
> | **MLP-Periodic** | 0.3371     | 0.7759     | 0.7637     | 0.9677     | 0.8595     | **0.5788** | 0.9666     | 0.7463     | 0.4277     | **0.2991** | 0.4353     | 0.9058     | 0.0246     |
> | **MLP-RaBEL**    | **0.8090** | **0.8495** | **0.8377** | **0.9692** | 0.8678     | 0.5758     | **0.9668** | **0.6781** | **0.3573** | 0.3101     | **0.4203** | 0.9043     | 0.0221     |
>
> 2. Additional Transformer-based Experiments
>
> Furthermore, to demonstrate the generalization capability of our method across different architectures, we have included results using a Transformer backbone. These results follow the MLP section in the revised manuscript.
>
> **Table 2: Classification Results on BCCO-REG**
>
> | **Embedding**         | **AUC (↑)** | **Acc. (↑)** | **F1 (↑)** |
> | --------------------- | ----------- | ------------ | ---------- |
> | Transformer+MLP       | 83.52       | 76.82        | 66.57      |
> | Transformer+Periodic  | 83.88       | 77.80        | 68.65      |
> | Transformer+PLE       | 84.66       | 77.68        | 67.74      |
> | **Transformer+RaBEL** | **85.04**   | **77.99**    | **69.01**  |
>
> **Table 3: Regression Results on BCCO-REG**
>
> | **Embedding**         | **R$^2$ (↑)** | **RMSE (↓)** |
> | --------------------- | ------------- | ------------ |
> | Transformer+MLP       | 0.7731        | 0.4043       |
> | Transformer+Periodic  | 0.6859        | 0.4321       |
> | Transformer+PLE       | 0.7410        | 0.4216       |
> | **Transformer+RaBEL** | **0.7792**    | **0.3964**   |
>
> We believe these comprehensive quantitative results robustly validate the efficacy of our proposed method.

---

> ### Author Response · Authors · 2025-11-26
>
> # Q-T 6: Experimental setup and baseline tuning.
> We appreciate the detailed inquiries regarding our experimental protocol. Below we address your questions point-by-point:
>
> - **Setup & Normalization:** We utilized ensemble settings and applied **Quantile transformation combined with SVD** for data preprocessing.
> - **Ranking:** We calculated the rank of each method on every individual dataset and reported the **mean rank** across all datasets.
> - **HPO & TabArena Divergence:** Regarding the performance of RealMLP and the divergence from official benchmarks, our initial setup followed standard **Tabular Foundation Model (TFM)** protocols, which prioritize zero-shot or fixed-configuration evaluations to assess generalization. Traditional deep learning baselines (like RealMLP) often require extensive tuning, and our initial use of default search spaces via the TALENT interface resulted in suboptimal performance for them.
>
> # Q-T 7: RaBEL Hyperparameter
> Hyperparameters and Sensitivity of RaBEL
> We thank the reviewer for pointing this out. We have added a comprehensive ablation study on the RaBEL embedding module in **Appendix A.4 (Lines 682-700)** to clarify the hyperparameter choices and their sensitivity.
>
> **1. Main Hyperparameters & Settings Used:**
> Regarding your specific question about $M$ (the number of centers), we refer to this as `n_kernels` in our experiments. In our reported results, we used **n_kernels=64** and **$\sigma=1.0$**, with **orthogonal initialization**, as these settings yielded the best performance.
>
> **2. Sensitivity Analysis:**
> As shown in the tables below, MiniX is relatively robust to hyperparameter variations, though proper tuning brings observable gains:
> * **Number of Kernels ($M$):** Performance improves as $M$ increases from 16 to 64.
> * **Initialization:** Orthogonal initialization significantly outperforms Xavier and Kaiming methods.
> * **Sigma ($\sigma$):** A fixed $\sigma=1.0$ works best; learnable $\sigma$ did not provide additional benefits.
>
> **Detailed Ablation Results:**
>
> | **Dim** | **AUC** | **Acc.** | **F1** | | **n kernels** | **AUC** | **Acc.** | **F1** |
> | :--- | :---: | :---: | :---: | :---: | :--- | :---: | :---: | :---: |
> | 16 | 84.51 | 77.68 | 68.32 | | 16 | 84.20 | 77.12 | 68.53 |
> | 32 | **85.17** | **77.95** | **69.12** | | 32 | 84.76 | 77.68 | 68.88 |
> | 64 | 84.78 | 77.82 | 68.71 | | 64 | **85.19** | **77.95** | **69.08** |
>
> | $\sigma$ | **AUC** | **Acc.** | **F1** | | **Init Method** | **AUC** | **Acc.** | **F1** |
> | :--- | :---: | :---: | :---: | :---: | :--- | :---: | :---: | :---: |
> | 0.5 | 84.66 | 77.62 | 68.85 | | Orthogonal | **89.03** | **77.67** | **68.95** |
> | 1.0 | **84.75** | **77.93** | **69.01** | | Xavier | 84.80 | 77.57 | 68.53 |
> | 2.0 | 84.67 | 77.44 | 68.04 | | Kaiming | 84.68 | 77.44 | 68.27 |
>
> | **Kernel Form** | **AUC** | **Acc.** | **F1** | | **$\sigma$ Mode** | **AUC** | **Acc.** | **F1** |
> | :--- | :---: | :---: | :---: | :---: | :--- | :---: | :---: | :---: |
> | Random | 84.43 | 77.67 | 68.55 | | Learn | 84.65 | 77.59 | 68.53 |
> | Uniform | **84.99** | **77.93** | **69.07** | | Fixed | **84.81** | **77.93** | **69.02** |
>
> **New Experiments (Rigorous HPO & TabM):** To ensure a fair comparison, we conducted a comprehensive re-evaluation with **rigorous per-dataset HPO** for RealMLP and added **TabM** as requested.
>
> - **Classification (Updated Tables 5-6):** Even against these heavily tuned baselines, MiniX remains highly competitive, ranking **2nd** overall (surpassed only by the much larger LimiX-16M).
> - **Regression:** While tuned specialists (RealMLP, TabM) show marginal gains, MiniX continues to outperform other foundation models (e.g., TabPFN-v2), confirming that our results reflect genuine model robustness rather than weak baselines.
>
> # Q-T 8: Experiments on Large Datasets
>
> To ensure a **fair and consistent comparison**, we adopted the same dataset filtering criteria as LimiX. This restriction was necessary to guarantee that **all** foundation model baselines could be evaluated on **every** dataset in the benchmark, avoiding the unfair exclusion of baselines that cannot scale to larger data.

---

> ### Author Response · Authors · 2025-11-26
>
> # Q-T 9: The flatten dimension of rank calculation
> Unlike Traditional Tabular Models (TTMs), Tabular Foundation Models (TFMs) employ a distinct processing paradigm. Given an input table of shape $(s, f)$, while both approaches may initially project the data into a high-dimensional space $(s, f, d)$, traditional models typically flatten this tensor into $(s, f \times d)$ for subsequent processing. In this context, calculating the rank of the $(s, f \times d)$ representation is intuitive, as the model operates on a sample-level basis.
>
> In contrast, TFMs preserve the multidimensional structure $(s, f, d)$ throughout the entire pipeline. Specifically, they utilize shapes of $(\dots, f, d)$ for feature attention and $(\dots, s, d)$ for sample attention, effectively maintaining a **cell-level** representation. Consequently, we argue that calculating the rank of the flattened $(s \times f, d)$ matrix is a more appropriate metric for analyzing Tabular Foundation Models.
>
> # Q-P 1: Section 4.2 clarity and RaBEL validation.
> We sincerely appreciate your detailed feedback on Section 4.2. We have substantially rewritten this section to improve clarity and eliminate redundancy, particularly regarding the discussion of duplicate text. Furthermore, to address the request for experimental validation, we have added a comprehensive hyperparameter sensitivity analysis for RaBEL in our **Response to W7** and we put it in Appendix A.4 Line 682-701. This additional study empirically justifies our design choices and demonstrates the robustness of the proposed embedding method.
>
> # Q-P 2: Definition of "Energy" (Line 163)
> Your assumption is correct. In this context, **"full energy"** is defined as the **sum of squared singular values** ($\sum \sigma_i^2$). Consequently, the metric **Rank@99%** denotes the minimum number of singular values required for their cumulative sum of squares to exceed 99% of the total sum.
>
> # Q-P 3: Title revision suggestion
> We appreciate your insightful suggestion. Upon re-evaluating our content, we consider both the RBA mechanism and the RaBEL embedding to be critical and equal contributions to the MiniX architecture. To accurately reflect this dual focus and the broader impact of our work, we have updated the title to: "MiniX: Mitigating Low-Rank Collapse and Attention Bottlenecks in Tabular Foundation Models". This revision highlights our solutions to both the low-rank problem and structural inefficiencies.
>
> # Q-P 4: Request for implementation details
> We have updated the manuscript to explicitly include experimental setups and implementation details in the relevant sections. Specifically, we clarify that:Data Generation: We construct our pre-training corpus using hierarchical Structural Causal Models (SCMs), following protocols established in the PFN series and LimiX. In each episode, synthetic data is drawn from random DAGs and functional mechanisms.Model Architecture: The MiniX backbone consists of a 12-block Transformer. Distinctively, each block follows a Sample-Attention $\rightarrow$ FFN $\rightarrow$ Feature-Attention sequence, explicitly modeling both inter-sample and intra-feature dependencies. The model is configured with a hidden dimension of $d_{\text{model}} = 96$ and $H=6$ attention heads.

---

### Official Review · Reviewer_bkwk · 2025-10-29

**Soundness:** 2
**Presentation:** 2
**Contribution:** 2
**Rating:** 2
**Confidence:** 4

**Summary:**

The submission proposes two architectural design choices for tabular foundational models (TFMs) that allow to achieve the lightweight model which is comparable to the state-of-the-art in terms of predictive accuracy.

**Strengths:**

The paper investigates the design choices for tabular foundation models, which is an active research direction. In my subjective opinion, the field is currently at the point where we should firstly focus on improving predictive accuracy rather than efficiency, since for modern hardware the existing TFMs are quite lightweight. But the paper findings indeed can be interesting to the subset of community working on the TFMs architectures.

**Weaknesses:**

(1) I am not fully satisfied with the positioning of the paper. Only one of two equally important contributions is presented in the title, why? For instance, Table 4 shows that on TabZilla RBA is more important than Rabel.

(2) The submission makes several claims that are not supported:

  (2.1) line 43 - "low rank of activations impedes gradient flow" - I did not find the proof in the paper.

   (2.2) line 73 - "strengthening gradient signals throughout the stack and improving parameter utilization" - I did not find the proof in the paper.

Without empirical confirmation each of these point is only a guess, intuition.

(3) The recent TabM model is not included in the comparison.

(4) From the description of new embeddings, it is not clear what are the main hyperparameters of the scheme and the sensitivity of Minix to them. I assume that the important hyperparameter is M (the number of centers) but I did not find any instructions how to choose its value and what value was used in the experiments.

(5) The authors do not report the advantage of Minix in terms of runtime efficiency (wall-clock time). The advantage in terms of memory in my opinion is less interesting since tabular foundational models are quite compact compared to models from other domains (CV, NLP).

(6) From the current set of experiments I do not buy the claim "The hidden states output by transformer layers tend to be low-rank, especially early in the network. This could severely decrease the expressivity of the network, leading to potential performance degradation." I appreciate the results from Table 1 but it demonstrates only the redundancy of TabPFNv2, which could be exploited to achieve higher efficiency. From the submission, I do not see any evidence that low-rank activations can be harmful for performance in terms of predictive accuracy.

**Questions:**

(1) Please, address my concerns in the Weaknesses section.
(2) Why the authors did not investigate the advantage of their embeddings beyond the context of tabular foundational models, for instance with the simplest MLP? Would they work better than MLP-PLR? In my opinion, such experiments would highlight the advantage of RaBEL better. Or is there some special chemistry between the RaBEL and TabPFN-like models? I appreciate the experiments from section 5.1 but it would be more interesting to extend them to real problems.

---

> ### Author Response · Authors · 2025-11-26
>
> # W1: Positioning and Title Refinement
> We appreciate the suggestion to elevate the positioning of the RBA module. We agree that RBA is critical to our performance (as evidenced by Table 4) and warrants equal prominence alongside RaBEL.
>
> **Actions Taken:**
> * **Title Update:** We have revised the title to explicitly reflect this focus: **"MiniX: Mitigating Low-Rank Collapse and Attention Bottlenecks in Tabular Foundation Models."**
> * **Expanded Analysis:** We enhanced the discussion and analysis of the RBA module to align better with our empirical findings (**Main Text: Lines 374-380**; **Appendix: Lines 637-643**).
>
> # W2: Insufficient Experimental Support for Claims on Low Rank and Scaling
> Thank the reviewer for this meticulous comment. We wish to clarify that our statement regarding low rank "hindering gradient flow" and "blunting scaling benefits" was primarily motivated by established Transformer theory [1, 2] rather than solely by our empirical results.
>
> Dong et al. [1] proved that without residual connections, attention rank decays doubly exponentially with depth, limiting the expressivity gains from scaling. Furthermore, Noci et al. [2] linked rank collapse to vanishing gradients and poor trainability.
> We agree that while we provided indirect evidence (rank comparison in Fig. 1), we did not include direct scaling curves to empirically verify this causal link. We acknowledge our original phrasing was too strong.
>
> We have revised **Lines 44–48** to accurately frame the low-rank issue as our *theoretical motivation* supported by literature, rather than an empirically proven conclusion of this work.
>
> **References:**
>
> [1] Dong et al., *ICML* 2021. Attention is not all you need: Pure attention loses rank doubly exponentially with depth.
>
> [2] Noci et al., *NeurIPS* 2022. Signal propagation in transformers: Theoretical perspectives and the role of rank collapse.
>
> # W3: Lack of TabM result
> We have conducted rigorous Hyperparameter Optimization (HPO) for **TabM** and have updated the results across all relevant tables in the revised manuscript.
>
> # W4: Hyperparameters and Sensitivity of RaBEL
> We thank the reviewer for pointing this out. We have added a comprehensive ablation study on the RaBEL embedding module in **Appendix A.4 (Lines 682-700)** to clarify the hyperparameter choices and their sensitivity.
>
> **1. Main Hyperparameters & Settings Used:**
> Regarding your specific question about $M$ (the number of centers), we refer to this as `n_kernels` in our experiments. In our reported results, we used **n_kernels=64** and **$\sigma=1.0$**, with **orthogonal initialization**, as these settings yielded the best performance.
>
> **2. Sensitivity Analysis:**
> As shown in the tables below, MiniX is relatively robust to hyperparameter variations, though proper tuning brings observable gains:
> * **Number of Kernels ($M$):** Performance improves as $M$ increases from 16 to 64.
> * **Initialization:** Orthogonal initialization significantly outperforms Xavier and Kaiming methods.
> * **Sigma ($\sigma$):** A fixed $\sigma=1.0$ works best; learnable $\sigma$ did not provide additional benefits.
>
> **Detailed Ablation Results:**
>
> | **Dim** | **AUC** | **Acc.** | **F1** | | **n kernels** | **AUC** | **Acc.** | **F1** |
> | :--- | :---: | :---: | :---: | :---: | :--- | :---: | :---: | :---: |
> | 16 | 84.51 | 77.68 | 68.32 | | 16 | 84.20 | 77.12 | 68.53 |
> | 32 | **85.17** | **77.95** | **69.12** | | 32 | 84.76 | 77.68 | 68.88 |
> | 64 | 84.78 | 77.82 | 68.71 | | 64 | **85.19** | **77.95** | **69.08** |
>
> | $\sigma$ | **AUC** | **Acc.** | **F1** | | **Init Method** | **AUC** | **Acc.** | **F1** |
> | :--- | :---: | :---: | :---: | :---: | :--- | :---: | :---: | :---: |
> | 0.5 | 84.66 | 77.62 | 68.85 | | Orthogonal | **89.03** | **77.67** | **68.95** |
> | 1.0 | **84.75** | **77.93** | **69.01** | | Xavier | 84.80 | 77.57 | 68.53 |
> | 2.0 | 84.67 | 77.44 | 68.04 | | Kaiming | 84.68 | 77.44 | 68.27 |
>
> | **Kernel Form** | **AUC** | **Acc.** | **F1** | | **$\sigma$ Mode** | **AUC** | **Acc.** | **F1** |
> | :--- | :---: | :---: | :---: | :---: | :--- | :---: | :---: | :---: |
> | Random | 84.43 | 77.67 | 68.55 | | Learn | 84.65 | 77.59 | 68.53 |
> | Uniform | **84.99** | **77.93** | **69.07** | | Fixed | **84.81** | **77.93** | **69.02** |

---

> ### Author Response · Authors · 2025-11-26
>
> # W5: Inference Speed
> We evaluate inference efficiency on a synthetic dataset (900 samples, 60 features), averaging three runs on an AMD EPYC 9354 CPU and NVIDIA RTX 4090 GPU (Table 1). MiniX achieves 171.40 ms on GPU, outperforming TabPFN-v2 by $\approx$2$\times$ and TabICL by $>10\times$. Additionally, MiniX demonstrates superior CPU efficiency compared to heavy baselines like Mitra, confirming its practical deployability.
>
> | Model     |   CPU (ms)   |  GPU (ms)  |
> | :-------- | :----------: | :--------: |
> | TabPFN-v2 |   51950.08   |   352.60   |
> | LimiX     |   68447.99   |   368.08   |
> | TabICL    |   22161.85   |  1749.61   |
> | Mitra     |  124453.05   |  5766.25   |
> | MiniX     | **17257.34** | **171.40** |
>
> # W6: Claims on Low-Rank and Performance Degradation
> We agree with the reviewer that low-rank activations indicate redundancy, which can indeed be exploited for efficiency. However, our results suggest that **alleviating this low-rank bottleneck unlocks significant performance gains**, implying that the low rank was indeed limiting the model's expressivity. As discussed in our **Response to W2**, while this view is supported by established Transformer literature, our empirical findings provide practical evidence: by designing MiniX to mitigate rank collapse (especially in early layers), we achieve superior predictive accuracy compared to the low-rank baseline (TabPFN-v2). This correlation strongly suggests that addressing the low-rank issue contributes to better model performance.

---

> ### Author Response · Authors · 2025-11-26
>
> # Q2: Comparisons between RaBEL and other numerical embedding methods
>  Initially, we designed the toy experiments to provide an **intuitive visualization** of how different embedding methods impact the representation space. However, we agree that qualitative visualizations alone are insufficient and that rigorous quantitative benchmarks provide stronger evidence for the method's effectiveness.
>
> To address this, we have significantly expanded our evaluation to include comprehensive quantitative experiments.
>
> 1. Additional MLP-based Experiments
>
> We have supplemented the paper with extensive MLP-based experimental results. These updates have been incorporated into the main text in Lines 290–323. As shown in the table below, our method (RaBEL) consistently outperforms other baselines across the majority of datasets.
>
> **Table 1: Performance comparison on various datasets using MLP backbone.**
>
> | **Metric1**       | **GC (↑)** | **CP (↑)** | **AC (↑)** | **CB (↑)** | **UK (↑)** | **BN (↑)** | **MA (↑)** | **CD (↑)** | **MH (↑)** | **CC (↑)** | **HS (↑)** | **SC (↑)** | **MV (↑)** |
> | ---------------- | ---------- | ---------- | ---------- | ---------- | ---------- | ---------- | ---------- | ---------- | ---------- | ---------- | ---------- | ---------- | ---------- |
> | **MLP**          | 0.7537     | 0.8092     | 0.6560     | 0.8319     | 0.9850     | 0.7520     | **0.8896** | 0.4476     | 0.7811     | 0.6158     | 0.7497     | 0.1799     | **0.9999** |
> | **MLP-MLP**      | 0.8984     | 0.8496     | 0.8398     | 0.8924     | **0.9863** | 0.7518     | 0.8577     | 0.4768     | 0.8618     | 0.8947     | 0.8375     | 0.1749     | 0.9990     |
> | **MLP-PLE**      | 0.7393     | 0.8031     | 0.8262     | 0.8817     | 0.8416     | 0.7389     | 0.8609     | 0.2969     | 0.8281     | 0.8576     | 0.8279     | **0.1815** | 0.9730     |
> | **MLP-Periodic** | 0.7817     | 0.8895     | 0.8054     | 0.8926     | 0.9821     | **0.7586** | 0.8625     | 0.4095     | 0.8306     | **0.9068** | 0.8286     | 0.1786     | 0.9994     |
> | **MLP-RaBEL**    | **0.9831** | **0.9582** | **0.9061** | **0.8979** | 0.9677     | 0.7580     | 0.8789     | **0.5124** | **0.8818** | 0.8998     | **0.8401** | 0.1813     | 0.9995     |
>
>
> | **Metric2**       | **GC (↑)** | **CP (↑)** | **AC (↑)** | **CB (↑)** | **UK (↑)** | **BN (↑)** | **MA (↑)** | **CD (↓)** | **MH (↓)** | **CC (↓)** | **HS (↓)** | **SC (↓)** | **MV (↓)** |
> | ---------------- | ---------- | ---------- | ---------- | ---------- | ---------- | ---------- | ---------- | ---------- | ---------- | ---------- | ---------- | ---------- | ---------- |
> | **MLP**          | 0.3483     | 0.6756     | 0.6750     | 0.9677     | **0.9008** | 0.5757     | 0.9611     | 0.7218     | 0.4862     | 0.6072     | 0.5259     | 0.9051     | **0.0119** |
> | **MLP-MLP**      | 0.5393     | 0.7224     | 0.7870     | **0.9692** | **0.9008** | 0.5766     | 0.9657     | 0.7025     | 0.3863     | 0.3178     | 0.4238     | 0.9078     | 0.0321     |
> | **MLP-PLE**      | 0.2584     | 0.6890     | 0.7750     | 0.9677     | 0.5868     | 0.5596     | 0.9654     | 0.8143     | 0.4308     | 0.3697     | 0.4361     | **0.9042** | 0.1642     |
> | **MLP-Periodic** | 0.3371     | 0.7759     | 0.7637     | 0.9677     | 0.8595     | **0.5788** | 0.9666     | 0.7463     | 0.4277     | **0.2991** | 0.4353     | 0.9058     | 0.0246     |
> | **MLP-RaBEL**    | **0.8090** | **0.8495** | **0.8377** | **0.9692** | 0.8678     | 0.5758     | **0.9668** | **0.6781** | **0.3573** | 0.3101     | **0.4203** | 0.9043     | 0.0221     |
>
> 2. Additional Transformer-based Experiments
>
> Furthermore, to demonstrate the generalization capability of our method across different architectures, we have included results using a Transformer backbone. These results follow the MLP section in the revised manuscript.
>
> **Table 2: Classification Results on BCCO-REG**
>
> | **Embedding**         | **AUC (↑)** | **Acc. (↑)** | **F1 (↑)** |
> | --------------------- | ----------- | ------------ | ---------- |
> | Transformer+MLP       | 83.52       | 76.82        | 66.57      |
> | Transformer+Periodic  | 83.88       | 77.80        | 68.65      |
> | Transformer+PLE       | 84.66       | 77.68        | 67.74      |
> | **Transformer+RaBEL** | **85.04**   | **77.99**    | **69.01**  |
>
> **Table 3: Regression Results on BCCO-REG**
>
> | **Embedding**         | **R$^2$ (↑)** | **RMSE (↓)** |
> | --------------------- | ------------- | ------------ |
> | Transformer+MLP       | 0.7731        | 0.4043       |
> | Transformer+Periodic  | 0.6859        | 0.4321       |
> | Transformer+PLE       | 0.7410        | 0.4216       |
> | **Transformer+RaBEL** | **0.7792**    | **0.3964**   |
>
> We believe these comprehensive quantitative results robustly validate the efficacy of our proposed method.

---

### Official Review · Reviewer_bJJG · 2025-11-01

**Soundness:** 2
**Presentation:** 2
**Contribution:** 1
**Rating:** 2
**Confidence:** 5

**Summary:**

This paper introduces RaBEL, a radial-basis embedding layer to address the low-rank bottleneck in early layers of tabular foundation models, and proposes a reordered bidirectional attention stack (sample-attention → FFN → feature-attention); it further presents MiniX, a 2M-parameter TFM.

**Strengths:**

1. The paper is relatively clear and easy to understand.
2. Analyzing the model from the rank perspective is interesting.
3. The proposed method achieves comparable performance on datasets (≤50,000 training samples, ≤10 target classes) as defined in the paper.

**Weaknesses:**

1. Since MiniX and LimiX share the same training setup, including their experimental protocols and optimization configurations, incorporating LimiX into comparative analyses—such as the experiment on "rank evolution across layers"—would strengthen the validity of conclusions. This addition would help clarify whether observed differences in performance or rank dynamics stem from architectural modifications (e.g., RaBEL vs. LimiX’s embedding) rather than training discrepancies, enhancing the rigor of the comparative framework.

2. The improvement of the model performance mainly comes from both RaBEL and RBA, yet the ablation experiment shows that the performance gain brought by RaBEL is not significantly higher than that of RBA. However, the paper focuses primarily on RaBEL. This mismatch between the core focus and experimental results may require a revision of the paper’s structure—either to adjust the emphasis on RaBEL’s differentiated value or to clarify why RaBEL is prioritized as the core contribution.

3. Regarding RBA: Compared with the module stacking order of TabPFN-v2, RBA not only rearranges the attention sequence (from "feature→sample" to "sample→feature") but also introduces an additional FFN. However, the paper lacks controlled experiments to disentangle the contributions of these two modifications. It remains unclear whether the performance improvement stems from the "order optimization" (aggregating sample-level statistics first before feature interaction) or the "nonlinear transformation of FFN" (compressing sample-wise information), which weakens the rigor of RBA’s design logic.

4. The number of model parameters is not as critical as implied. Among the compared baselines, TabICL—with the largest parameter count (27M)—exhibits the fastest inference speed, suggesting that inference speed is a more practical metric for tabular models.

5. The claim in Lines 41–44 (that low rank "hinders gradient flow" and "blunts the benefits of scaling deeper or wider backbones") lacks sufficient experimental support. While the paper indirectly supports the low-rank issue via rank comparisons between MiniX and TabPFN-v2 (Fig. 1), it does not provide direct evidence—such as verifying whether MiniX gains more relative performance when scaled up (e.g., deeper layers, more attention heads) compared to TabPFN-v2. This makes the claim more speculative than empirically grounded.

6. There is ambiguity about whether "RaBEL" and "ReBEL" (mentioned later in experiments like Fig. 2 and Tab. 4) refer to the same component. The paper does not clarify if "ReBEL" is a typo or a modified version of RaBEL. This inconsistency in terminology confuses readers’ understanding of the module design and requires standardization with explicit definitions.

7. The rationale for limiting datasets to ≤50,000 samples is unexplained. This criterion clearly favors small-sample and excludes common practical scenarios for tabular data (e.g., large-sample datasets with millions of instances or multi-class tasks), making it impossible to verify the model’s generalization to broader use cases. Additionally, the paper does not compare the distribution of filtered vs. unfiltered datasets or explain if this criterion unfairly benefits MiniX.

8. The transparency and comparability of experimental settings are insufficient:
   - Details about baselines are missing: It is unclear whether other models (e.g., TabR, ModernNCA, RealMLP) underwent hyperparameter optimization (HPO) or used ensemble strategies. This makes it impossible to explain why some models perform worse than simple MLPs (poor performance may stem from inadequate tuning rather than inherent weaknesses).
   - Data integrity is lacking: The TabM method (available in TabArena) is excluded, and the paper does not specify whether results for unmentioned methods are reused from TabArena’s official benchmarks or retrained by the authors. Inconsistent experimental protocols (e.g., training epochs, validation strategies) undermine result comparability.
   - Result traceability is unclear: It is not stated whether the overall results follow TabArena’s evaluation pipeline; any discrepancies (e.g., data splitting, metric calculation details) would distort cross-model comparisons.

9. Comparisons between RaBEL and other numerical embedding methods (e.g., -PLE, -PLR) are missing, leaving RaBEL’s differentiated advantages unproven. Additionally, the paper does not test whether RaBEL can yield similar performance improvements when integrated into other tabular deep learning architectures. Of course, this is merely a suggestion and not considered a major issue.

**Questions:**

Refer to the Weaknesses.

---

> ### Author Response · Authors · 2025-11-26
>
> # W1: Rank comparison with LimiX
> We thank you for highlighting this. To isolate the effects of training settings and parameter counts, we trained a 2M-parameter baseline incorporating RBA to directly compare the embedding rank changes before and after adding RaBEL. Since RaBEL is applied at the input layer, its impact is most observable in the shallow layers. Therefore, we compared the average rank across the first three layers for both models. The results are presented below and have been included in Appendix A.7 (Lines 1192-1196):
>
> | **Model**   | **Numerical Rank**  | **Rank@99%**        | **Rank@95%**        |
> | ----------- | ------------------- | ------------------- | ------------------- |
> | Baseline-2M | 58.41               | 13.94               | 6.73                |
> | MiniX       | **78.62** (+34.60%) | **25.35** (+81.98%) | **12.31** (+83.18%) |
>
> # W2: Lack the analysis of RBA
> We designed a new toy experiment to compare two architectures: SNF (sample attention $\to$ feed-forward network $\to$ feature attention) and FSN (feature attention $\to$ sample attention $\to$ feed-forward network), using a synthetic dataset generated from a Directed Acyclic Graph (DAG). Our analysis reveals that while the FSN baseline is dominated by self-attention, SNF accurately assigns high attention scores to the direct causal features of the target, confirming its superior ability to model complex feature interactions. Visualizations of these results are provided in Appendix A.3 (Lines 648-671).
>
> # W3: Clarification on RBA architecture.
> We apologize for the confusion and would like to clarify that RBA does not introduce any additional FFN modules. It is purely a structural reordering of the existing components. Specifically:The original design (TabPFN-v2) follows an FSN sequence: Feature Attention $\rightarrow$ Sample Attention $\rightarrow$ Feed-Forward Network.Our RBA design rearranges this into an SNF sequence: Sample Attention $\rightarrow$ Feed-Forward Network $\rightarrow$ Feature Attention.Therefore, the improvements stem from the optimized information flow rather than increased model capacity or extra modules. We have revised Section 4.3 to explicitly describe this reordering and eliminate any ambiguity regarding the architectural composition.
>
> # W4: Inference Speed
> We evaluate inference efficiency on a synthetic dataset (900 samples, 60 features), averaging three runs on an AMD EPYC 9354 CPU and NVIDIA RTX 4090 GPU (Table 1). MiniX achieves 171.40 ms on GPU, outperforming TabPFN-v2 by $\approx$2$\times$ and TabICL by $>10\times$. Additionally, MiniX demonstrates superior CPU efficiency compared to heavy baselines like Mitra, confirming its practical deployability.
>
> | Model     |   CPU (ms)   |  GPU (ms)  |
> | :-------- | :----------: | :--------: |
> | TabPFN-v2 |   51950.08   |   352.60   |
> | LimiX     |   68447.99   |   368.08   |
> | TabICL    |   22161.85   |  1749.61   |
> | Mitra     |  124453.05   |  5766.25   |
> | MiniX     | **17257.34** | **171.40** |
>
> # W5: Insufficient Experimental Support for Claims on Low Rank and Scaling
>
> We thank the reviewer for this insightful comment. We wish to clarify that our statement regarding low rank "hindering gradient flow" and "blunting scaling benefits" was primarily motivated by established Transformer theory [1, 2] rather than solely by our empirical results.
>
> Dong et al. [1] proved that without residual connections, attention rank decays doubly exponentially with depth, limiting the expressivity gains from scaling. Furthermore, Noci et al. [2] linked rank collapse to vanishing gradients and poor trainability.
> We agree that while we provided indirect evidence (rank comparison in Fig. 1), we did not include direct scaling curves to empirically verify this causal link. We acknowledge our original phrasing was too strong.
>
> We have revised **Lines 44–48** to accurately frame the low-rank issue as our *theoretical motivation* supported by literature, rather than an empirically proven conclusion of this work.
>
> **References:**
>
> [1] Dong et al., *ICML* 2021. Attention is not all you need: Pure attention loses rank doubly exponentially with depth.
>
> [2] Noci et al., *NeurIPS* 2022. Signal propagation in transformers: Theoretical perspectives and the role of rank collapse.
>
> # W6: Inconsistency in Terminology ("RaBEL" vs. "ReBEL")
> We sincerely apologize for the confusion caused by this oversight. We confirm that **"ReBEL" is indeed a typographical error** and refers to the exact same component as **RaBEL**. There is no modification or variation between the two; they are intended to denote the same module.  We have thoroughly proofread the manuscript and standardized all terminology to **"RaBEL"**. Specifically, we have corrected this content in the revised version to ensure consistency and clarity.
>
> Due to character limits, we have included the rest of the results in the subsequent comment.

---

> > ### Author Response · Authors · 2025-11-26
> >
> > # W7:  Justification for Dataset Selection Criteria ($\le$ 50k samples)
> > We clarify that the 50,000-sample threshold was selected to ensure a fair and feasible comparison with state-of-the-art Tabular Foundation Models (TFMs), rather than to cherry-pick favorable scenarios.
> >
> > * **Fair Benchmarking:** Existing TFMs have strict inference constraints due to context limitations. TabPFN-v2 is restricted to $\le$ 10,000 samples, and LimiX to $\le$ 50,000. We adopted the LimiX threshold to evaluate at the maximum scale feasible for side-by-side comparison.
> > * **Feasibility & OOM:** This criterion does not unfairly favor MiniX. Extending beyond 50k samples would render comparative analysis impossible, as baselines would likely fail due to Out-of-Memory (OOM) errors or prohibitive inference times.
> > * **Reproducibility:** Adhering to this limit aligns with standard practices in current TFM literature and ensures experimental reproducibility on standard hardware resources.
> >
> > # W8: Transparency of Baselines, HPO, and Data Protocols
> >
> > We thank the reviewer for emphasizing the importance of robust baselines.
> > * **Clarification on Protocols:** Our initial setup aligned with standard Tabular Foundation Model (TFM) evaluation, which typically assesses zero-shot generalization capabilities rather than per-dataset optimization. We relied on default TALENT benchmark configurations, which we acknowledge may have underrepresented the peak performance of traditional deep learning models.
> > * **New Experiments (Rigorous HPO):** To address this, we conducted a comprehensive re-evaluation with extensive per-dataset Hyperparameter Optimization (HPO) for deep learning baselines (RealMLP) and added **TabM** to the comparison. All models were rigorously retrained using a standardized pipeline.
> > * **Updated Results (Tables 5-6, App. A.5):**
> >     * **Classification:** MiniX remains highly competitive against these heavily tuned baselines, ranking **2nd** only to the significantly larger LimiX-16M.
> >     * **Regression:** While tuned specialists (RealMLP, TabM) achieve marginal gains, MiniX consistently outperforms other foundation models (e.g., TabPFN-v2), confirming that its performance is a result of robust design rather than weak baselines.

---

> ### Author Response · Authors · 2025-11-26
>
> # W9: Comparisons between RaBEL and other numerical embedding methods
>  Initially, we designed the toy experiments to provide an **intuitive visualization** of how different embedding methods impact the representation space. However, we agree that qualitative visualizations alone are insufficient and that rigorous quantitative benchmarks provide stronger evidence for the method's effectiveness.
>
> To address this, we have significantly expanded our evaluation to include comprehensive quantitative experiments.
>
> 1. Additional MLP-based Experiments
>
> We have supplemented the paper with extensive MLP-based experimental results. These updates have been incorporated into the main text in Lines 290–323. As shown in the table below, our method (RaBEL) consistently outperforms other baselines across the majority of datasets.
>
> **Table 1: Performance comparison on various datasets using MLP backbone.**
>
> | **Metric1**       | **GC (↑)** | **CP (↑)** | **AC (↑)** | **CB (↑)** | **UK (↑)** | **BN (↑)** | **MA (↑)** | **CD (↑)** | **MH (↑)** | **CC (↑)** | **HS (↑)** | **SC (↑)** | **MV (↑)** |
> | ---------------- | ---------- | ---------- | ---------- | ---------- | ---------- | ---------- | ---------- | ---------- | ---------- | ---------- | ---------- | ---------- | ---------- |
> | **MLP**          | 0.7537     | 0.8092     | 0.6560     | 0.8319     | 0.9850     | 0.7520     | **0.8896** | 0.4476     | 0.7811     | 0.6158     | 0.7497     | 0.1799     | **0.9999** |
> | **MLP-MLP**      | 0.8984     | 0.8496     | 0.8398     | 0.8924     | **0.9863** | 0.7518     | 0.8577     | 0.4768     | 0.8618     | 0.8947     | 0.8375     | 0.1749     | 0.9990     |
> | **MLP-PLE**      | 0.7393     | 0.8031     | 0.8262     | 0.8817     | 0.8416     | 0.7389     | 0.8609     | 0.2969     | 0.8281     | 0.8576     | 0.8279     | **0.1815** | 0.9730     |
> | **MLP-Periodic** | 0.7817     | 0.8895     | 0.8054     | 0.8926     | 0.9821     | **0.7586** | 0.8625     | 0.4095     | 0.8306     | **0.9068** | 0.8286     | 0.1786     | 0.9994     |
> | **MLP-RaBEL**    | **0.9831** | **0.9582** | **0.9061** | **0.8979** | 0.9677     | 0.7580     | 0.8789     | **0.5124** | **0.8818** | 0.8998     | **0.8401** | 0.1813     | 0.9995     |
>
>
> | **Metric2**       | **GC (↑)** | **CP (↑)** | **AC (↑)** | **CB (↑)** | **UK (↑)** | **BN (↑)** | **MA (↑)** | **CD (↓)** | **MH (↓)** | **CC (↓)** | **HS (↓)** | **SC (↓)** | **MV (↓)** |
> | ---------------- | ---------- | ---------- | ---------- | ---------- | ---------- | ---------- | ---------- | ---------- | ---------- | ---------- | ---------- | ---------- | ---------- |
> | **MLP**          | 0.3483     | 0.6756     | 0.6750     | 0.9677     | **0.9008** | 0.5757     | 0.9611     | 0.7218     | 0.4862     | 0.6072     | 0.5259     | 0.9051     | **0.0119** |
> | **MLP-MLP**      | 0.5393     | 0.7224     | 0.7870     | **0.9692** | **0.9008** | 0.5766     | 0.9657     | 0.7025     | 0.3863     | 0.3178     | 0.4238     | 0.9078     | 0.0321     |
> | **MLP-PLE**      | 0.2584     | 0.6890     | 0.7750     | 0.9677     | 0.5868     | 0.5596     | 0.9654     | 0.8143     | 0.4308     | 0.3697     | 0.4361     | **0.9042** | 0.1642     |
> | **MLP-Periodic** | 0.3371     | 0.7759     | 0.7637     | 0.9677     | 0.8595     | **0.5788** | 0.9666     | 0.7463     | 0.4277     | **0.2991** | 0.4353     | 0.9058     | 0.0246     |
> | **MLP-RaBEL**    | **0.8090** | **0.8495** | **0.8377** | **0.9692** | 0.8678     | 0.5758     | **0.9668** | **0.6781** | **0.3573** | 0.3101     | **0.4203** | 0.9043     | 0.0221     |
>
> 2. Additional Transformer-based Experiments
>
> Furthermore, to demonstrate the generalization capability of our method across different architectures, we have included results using a Transformer backbone. These results follow the MLP section in the revised manuscript.
>
> **Table 2: Classification Results on BCCO-REG**
>
> | **Embedding**         | **AUC (↑)** | **Acc. (↑)** | **F1 (↑)** |
> | --------------------- | ----------- | ------------ | ---------- |
> | Transformer+MLP       | 83.52       | 76.82        | 66.57      |
> | Transformer+Periodic  | 83.88       | 77.80        | 68.65      |
> | Transformer+PLE       | 84.66       | 77.68        | 67.74      |
> | **Transformer+RaBEL** | **85.04**   | **77.99**    | **69.01**  |
>
> **Table 3: Regression Results on BCCO-REG**
>
> | **Embedding**         | **R$^2$ (↑)** | **RMSE (↓)** |
> | --------------------- | ------------- | ------------ |
> | Transformer+MLP       | 0.7731        | 0.4043       |
> | Transformer+Periodic  | 0.6859        | 0.4321       |
> | Transformer+PLE       | 0.7410        | 0.4216       |
> | **Transformer+RaBEL** | **0.7792**    | **0.3964**   |
>
> We believe these comprehensive quantitative results robustly validate the efficacy of our proposed method.

---

### Official Review · Reviewer_STK2 · 2025-11-03

**Soundness:** 2
**Presentation:** 3
**Contribution:** 3
**Rating:** 6
**Confidence:** 2

**Summary:**

The paper provides an analysis of tabular foundation models through the perspective of low-rank collapse in their respresentation. This motivates a new, more efficient, but performant architecture with two new components - Radial Bassi Embedding Layer and revamped base PFN transformer layer structure. Authors provide intuitions, toy experiments and introspections into tabular foundation models as evidence plus they demonstrate the resulting model's (MiniX) performance empirically on a set of benchmarks (TabArena, Tabzilla and the OpenML CTR23 regression dataset suite) where MiniX lands second after only the current SoTA (as far as I personally know) tabular foundation model.

**Strengths:**

I find the paper fresh and useful for the literature on tabular foundation models. In my view the current state of affairs in this field is heavily in need of analysis and understanding. The paper contributes to our understanding a fair amount of new knowledge in my view.

My favorite takeaway from reading the paper  is that tabular foundation models are "appropriately" scaled in terms of parameter counts, or even a bit redundant at present. Contrary to the meta in other domains (e.g. LLMs and computer vision models getting into billions of parameters), tabular foundation models are different, and one can achieve close-to-SoTA results with a very small model. This overall feeling sways me towards accept, I belive this paper provides valuable insight to the community.

Result in Table 1 and Figure 1 are intriguing. We can see that current models hidden-state is heavily underutilized - this is an interesting and new piece of knowledge about ICL-based tabular foundation models.

A new feature embedding (RaBEL) is also interesting, It seems that it may be less sensitive to hyperparameters than prior work alternatives. But this, I belive, requires more testing and reporting. Another change (RBA) also seems notable, judging by the ablation in Table 4.

**Weaknesses:**

Despite me overall rating the paper positively and leaning towards accept. I think it has some downsides and problems that if addressed would make it better and make me more confident in my assessment.

1. The results reported in Table 2 and especially Table 3 are a bit too minimalistic. The paper provides only the aggregate scores and focuses mostly on the average AUC/R^2. When you take a closer look at the average rank, the MiniX model is near or below XGBoost on CTR23 for example. Providing as much and as granular as possible results would make the paper clearer and stronger in my view. I recommend to report individual dataset results and some form of confidence intervals or any other appropriate mean for a fairer comparison.
2. Regarding the analysis of the low-rank collapse in the representation. Why do you calculate the rank the way you do by flattening samples and features into one dimension. I think this should be explained more thoroughly. I can see an alternative strategy for computing the inner rank e.g. by flattening each individual object, instead of feature (e.g. $N \times M \times D$ into $N \times (M \cdot D)$. Also what about more granular rank statistics? E.g. Per token or per layer - the target corresponding token could be different from the other ones for example. Could you provide more insight into your thinking and maybe provide alternative rank stats?
3. The toy experiments part that introduced the new embedding scheme seems a bit less strong empirically. The only two examples are toy functions. I would've liked to see some more evidence for the proposed scheme (embeddings are also useful in non-foundation models, as can be seen in the original paper ["On Embeddings for Numerical Features in Tabular Deep Learning"](https://arxiv.org/abs/2203.05556) or in the recent RealMLP or TabM models. Demonstrating the efficacy of the proposed scheme in general would strengthen the claim that is made on the strength of RaBEL embeddings.
4. There is a statement made that same performance could be attained with a smaller model. As I outlined in strength - I like this finding in general. But the paper does not dive as deep into this point as it perfectly would in my view. First, the additional architectural improvements are at odds with this a bit. I guess what could imporve the paper is adding the baseline 2M model into the overall comparison (with all the rigor from point 1 I've mentioned). I believe it could be on par with the Mitra model, which is notable for proving the point of model redundancy and raises question on why there is such a difference? data? better hyperparameters? (exactly the missing analysis and ablation-style studies in the current tabular PFN world) Second, is the question whether scaling up MiniX would actually make the best model? This may be harder to address experimentally during the rebuttal period, I believe this should be at least discussed in conclusion.

**Questions:**

see weaknesses. Each point is an addressable question

---

> ### Author Response · Authors · 2025-11-26
>
> # W1: The results in Tab 2 and Tab 3 are too minimalistic
> We appreciate the reviewer’s suggestion to provide a more detailed performance breakdown. Due to the extensive number of datasets in our benchmarks, listing individual metrics for every dataset within the main text is spatially prohibitive. However, to address your concern and offer the requested granularity, we have introduced **waterfall plots** (see Appendix A.8 Figure 4 (Line 1228-1240) and Figure 6 (Line 1277-1289) )) for the **TabArena-CLS** and **CTR23** benchmarks.
>
> These plots visualize the head-to-head win/loss statistics of MiniX against **TabICL** **TabPFN-v2**, **XGBoost**, **CatBoost** and **Mitra**, providing a clear, fine-grained perspective on the exact number of datasets where MiniX outperforms these baselines.  Note that LimiX was excluded from this specific visualization to better highlight the comparative dynamics between MiniX and the other widely-used baselines. This analysis demonstrates that MiniX maintains a competitive edge on a significant portion of specific tasks.
>
> # W2: The flatten dimension of rank calculation
> Unlike Traditional Tabular Models (TTMs), Tabular Foundation Models (TFMs) employ a distinct processing paradigm. Given an input table of shape $(s, f)$, while both approaches may initially project the data into a high-dimensional space $(s, f, d)$, traditional models typically flatten this tensor into $(s, f \times d)$ for subsequent processing. In this context, calculating the rank of the $(s, f \times d)$ representation is intuitive, as the model operates on a sample-level basis.
>
> In contrast, TFMs preserve the multidimensional structure $(s, f, d)$ throughout the entire pipeline. Specifically, they utilize shapes of $(\dots, f, d)$ for feature attention and $(\dots, s, d)$ for sample attention, effectively maintaining a **cell-level** representation. Consequently, we argue that calculating the rank of the flattened $(s \times f, d)$ matrix is a more appropriate metric for analyzing Tabular Foundation Models.
>
> # W4: Discussion on 2M Baseline and Model Scaling
> In response to your query, we have supplemented our analysis by including comparison results against a **2M-parameter baseline model**. However, as noted in our response to W1, we can't displaying the full results across all datasets. Consequently, we have generated two **waterfall plots**—consistent with our previous visualizations—to illustrate the win/loss performance of the 2M baseline on the TabArena-CLS (classification) and CTR23 (regression) benchmarks ( see Appendix A.8 Figure 3 (Line 1213-1223) and Figure 5 (Line 1249-1261) .
>
> Regarding the scaling of **MiniX**, the limited duration of the rebuttal period prevented us from completing these computationally intensive experiments immediately. We are currently running these tests and will update the **discussion area** with the new results as soon as they are available.

---

> ### Author Response · Authors · 2025-11-26
>
> # W3: The results of toy experiment are weak and need for more evidence
>
> We sincerely appreciate the reviewer's feedback. Initially, we designed the toy experiments to provide an **intuitive visualization** of how different embedding methods impact the representation space. However, we agree that qualitative visualizations alone are insufficient and that rigorous quantitative benchmarks provide stronger evidence for the method's effectiveness.
>
> To address this, we have significantly expanded our evaluation to include comprehensive quantitative experiments.
>
> 1. Additional MLP-based Experiments
>
> We have supplemented the paper with extensive MLP-based experimental results. These updates have been incorporated into the main text in Lines 290–323. As shown in the table below, our method (RaBEL) consistently outperforms other baselines across the majority of datasets.
>
> **Table 1: Performance comparison on various datasets using MLP backbone.**
>
> | **Metric1**       | **GC (↑)** | **CP (↑)** | **AC (↑)** | **CB (↑)** | **UK (↑)** | **BN (↑)** | **MA (↑)** | **CD (↑)** | **MH (↑)** | **CC (↑)** | **HS (↑)** | **SC (↑)** | **MV (↑)** |
> | ---------------- | ---------- | ---------- | ---------- | ---------- | ---------- | ---------- | ---------- | ---------- | ---------- | ---------- | ---------- | ---------- | ---------- |
> | **MLP**          | 0.7537     | 0.8092     | 0.6560     | 0.8319     | 0.9850     | 0.7520     | **0.8896** | 0.4476     | 0.7811     | 0.6158     | 0.7497     | 0.1799     | **0.9999** |
> | **MLP-MLP**      | 0.8984     | 0.8496     | 0.8398     | 0.8924     | **0.9863** | 0.7518     | 0.8577     | 0.4768     | 0.8618     | 0.8947     | 0.8375     | 0.1749     | 0.9990     |
> | **MLP-PLE**      | 0.7393     | 0.8031     | 0.8262     | 0.8817     | 0.8416     | 0.7389     | 0.8609     | 0.2969     | 0.8281     | 0.8576     | 0.8279     | **0.1815** | 0.9730     |
> | **MLP-Periodic** | 0.7817     | 0.8895     | 0.8054     | 0.8926     | 0.9821     | **0.7586** | 0.8625     | 0.4095     | 0.8306     | **0.9068** | 0.8286     | 0.1786     | 0.9994     |
> | **MLP-RaBEL**    | **0.9831** | **0.9582** | **0.9061** | **0.8979** | 0.9677     | 0.7580     | 0.8789     | **0.5124** | **0.8818** | 0.8998     | **0.8401** | 0.1813     | 0.9995     |
>
>
> | **Metric2**       | **GC (↑)** | **CP (↑)** | **AC (↑)** | **CB (↑)** | **UK (↑)** | **BN (↑)** | **MA (↑)** | **CD (↓)** | **MH (↓)** | **CC (↓)** | **HS (↓)** | **SC (↓)** | **MV (↓)** |
> | ---------------- | ---------- | ---------- | ---------- | ---------- | ---------- | ---------- | ---------- | ---------- | ---------- | ---------- | ---------- | ---------- | ---------- |
> | **MLP**          | 0.3483     | 0.6756     | 0.6750     | 0.9677     | **0.9008** | 0.5757     | 0.9611     | 0.7218     | 0.4862     | 0.6072     | 0.5259     | 0.9051     | **0.0119** |
> | **MLP-MLP**      | 0.5393     | 0.7224     | 0.7870     | **0.9692** | **0.9008** | 0.5766     | 0.9657     | 0.7025     | 0.3863     | 0.3178     | 0.4238     | 0.9078     | 0.0321     |
> | **MLP-PLE**      | 0.2584     | 0.6890     | 0.7750     | 0.9677     | 0.5868     | 0.5596     | 0.9654     | 0.8143     | 0.4308     | 0.3697     | 0.4361     | **0.9042** | 0.1642     |
> | **MLP-Periodic** | 0.3371     | 0.7759     | 0.7637     | 0.9677     | 0.8595     | **0.5788** | 0.9666     | 0.7463     | 0.4277     | **0.2991** | 0.4353     | 0.9058     | 0.0246     |
> | **MLP-RaBEL**    | **0.8090** | **0.8495** | **0.8377** | **0.9692** | 0.8678     | 0.5758     | **0.9668** | **0.6781** | **0.3573** | 0.3101     | **0.4203** | 0.9043     | 0.0221     |
>
> 2. Additional Transformer-based Experiments
>
> Furthermore, to demonstrate the generalization capability of our method across different architectures, we have included results using a Transformer backbone. These results follow the MLP section in the revised manuscript.
>
> **Table 2: Classification Results on BCCO-REG**
>
> | **Embedding**         | **AUC (↑)** | **Acc. (↑)** | **F1 (↑)** |
> | --------------------- | ----------- | ------------ | ---------- |
> | Transformer+MLP       | 83.52       | 76.82        | 66.57      |
> | Transformer+Periodic  | 83.88       | 77.80        | 68.65      |
> | Transformer+PLE       | 84.66       | 77.68        | 67.74      |
> | **Transformer+RaBEL** | **85.04**   | **77.99**    | **69.01**  |
>
> **Table 3: Regression Results on BCCO-REG**
>
> | **Embedding**         | **R$^2$ (↑)** | **RMSE (↓)** |
> | --------------------- | ------------- | ------------ |
> | Transformer+MLP       | 0.7731        | 0.4043       |
> | Transformer+Periodic  | 0.6859        | 0.4321       |
> | Transformer+PLE       | 0.7410        | 0.4216       |
> | **Transformer+RaBEL** | **0.7792**    | **0.3964**   |
>
> We believe these comprehensive quantitative results robustly validate the efficacy of our proposed method.

---

### Meta-Review · Area_Chair_qzgq · 2026-01-06

**Summary:**

The paper provides a method to handle tabular data with a more computationally efficient model. The main idea is a new embedding scheme motivated by the problem of embeddings having low rank.
The sentiment in the reviews towards the overall idea and motivation of having more efficient models for tabular data is positive, and the overall results seem encouraging. Moreover, the observation that the “low-rank” problem is not handled efficiently by existing models seem to be appreciated (e.g. hNSy “I think the low-rank problem is crucial, and this finding is the most valuable result of the paper.”).
Despite the potential, the paper has several key issues mentioned in multiple reviews: The comparison against recent models is said to be incomplete (bJJG:  “Comparisons between RaBEL and other numerical embedding methods (e.g., -PLE, -PLR) are missing”, bkwk “The recent TabM model is not included in the comparison”). There is a need for more ablations and experiments evaluating different aspects of the model such as sensitivity to hyper-parameters, runtime efficiency (mentioned by bkwk and bJJG), lack of ablations on embeddings (hNSy). Additionally, more than one review mentions claims in the paper that should be better backed by experiments.
Finally, the clarity of the experimental section (details about the setup, etc) are mentioned to be too vague.

The authors provided a rebuttal addressing many of these issues by either answering questions or providing additional experiments. However, I found these to be somewhat rushed, (understandably, given the time limit of a rebuttal). More importantly, the scope of the required change is too large for a rebuttal. I think the paper has potential since the method does seem to improve over the baselines, and the high level arguments are said to be convincing. However, it is in too much of a raw shape to be accepted, even after the revisions of the rebuttal.

**Reviewer Concerns:**

As stated above, the concerns were only partially mitigated, as they required an extensive addition, that was not provided in the rebuttal. I believe that the concerns mostly remain, even after the rebuttal

**Reviewer Scores:**

Some of the reviews may have changed their score from 2 to 4, but I don't think anyone of the 3 reviewers providing a score of 2 would change their vote to (weak) accept.

---

### Decision · Program_Chairs · 2026-01-26

Reject